# A stationary phase-specific bacterial green light sensor for enhancing metabolite production

John T. Lazar[1], Daniel J. Haller [2], Abbas Ghaddar[3], Jae J. Kim[3], Kevin Yang[3], Sebastián M. Castillo-Hair [3], Andrew R. Gilmour[2], Ross Thyer [1,2] & Jeffrey J. Tabor [1,2,3,4] ✉

Genetically-encoded sensors are used to control protein and metabolite production in bacterial fermentations. However, these sensors are generally optimized for exponential growth rather than stationary phase where production occurs. Here, we find that our previously engineered *E. coli* green light sensor CcaSR, which functions robustly in exponential phase, fails in stationary phase due to spontaneous loss of an engineered chromophore biosynthetic pathway and accumulation of CcaS and CcaR. We optimize the genetic context and expression determinants of each component, resulting in a stable system named CcaSR$_{stat}$ that imposes little metabolic burden, exhibits low leakiness and an 80-fold green light response, and functions exclusively in stationary phase. We combine CcaSR$_{stat}$-driven enzyme expression with varied static and periodic illumination patterns to achieve high titers of the industrially-relevant phenylpropanoid *p*-Coumaric acid and demonstrate that these optimizations scale to benchtop bioreactor conditions. Finally, we use CcaSR$_{stat}$ to optimize the expression level of a co-transcribed multi-enzyme metabolic pathway encoding production of plant-derived betaxanthin family pigments. Stationary phase-optimized bacterial sensors should enhance fermentation productivity by enabling rapid interrogation of the impact of enzyme expression level and induction dynamics.

Metabolic engineering has several benefits over traditional catalysis for chemical production[1,2]. Here, microbes are genetically engineered to express heterologous enzymes that convert low-cost, renewable substrates into high-value therapeutics, fuels, and industrial precursors at near-ambient temperatures and pressures[3]. For example, researchers have expressed pathways with up to 30 heterologous enzymes in optimized microbial hosts to produce complex metabolites such as the cancer drug vinblastine[4] or the cannabinoid derivative cannabigerolic acid[5]. In another example, microbes have been engineered to convert low-value $CO_2$ streams into industrially-relevant

quantities of acetone and isopropanol, creating sustainable carbon-negative production processes[6].

Microbial chemical production traditionally occurs in fermentations characterized by two distinct phases. The first is an exponential growth phase where cells consume nutrients and divide rapidly to reach high total biomass. The second is a production phase where nutrients become limited, growth halts (i.e., cells reach stationary phase), and metabolic flux is diverted toward the desired product[7,8]. The capacity of microbes to produce heterologous molecules is higher in stationary phase than exponential phase because product synthesis

[1]Department of Chemical and Biomolecular Engineering, Rice University, Houston, TX, USA. [2]Ph.D. Program in Systems, Synthetic, and Physical Biology, Rice University, Houston, TX, USA. [3]Department of Bioengineering, Rice University, Houston, TX, USA. [4]Department of Biosciences, Rice University, Houston, TX, USA. ✉e-mail: jeff.tabor@rice.edu

does not compete with biomass accumulation[9]. This elevated production capacity combines with high total biomass to make stationary phase the condition under which the greatest quantities of molecules can be produced[9].

Heterologous enzyme expression is often controlled by inducible promoter systems. These systems generally comprise a transcription factor protein that is modulated by a chemical inducer such as isopropyl ß-D-1-thiogalactopyranoside (IPTG) or L-arabinose, and a target promoter that is regulated by the transcription factor. Inducible promoter systems can reduce unwanted exponential phase enzyme expression and enable control over the timing and level of enzyme expression to increase product titer and yield[10–16].

Despite their routine use in metabolic engineering, inducible promoter systems are typically optimized to function in exponential phase[14,17–20]. However, stable proteins such as transcription factors tend to accumulate in stationary phase[9,21,22] and elevated concentrations of transcriptional repressors (e.g., LacI, AraC) can lead to lower levels of target enzyme expression even in the presence of saturating inducer[23]. Conversely, increases in transcriptional activator abundance can lead to high leaky target gene expression that can result in unwanted toxicity[14]. Thus, optimizing expression of transcription factors or their accessory proteins in exponential phase could result in sub-optimal enzyme induction in the production phase of fermentations. Therefore, there is a need to optimize inducible promoter systems to perform robustly in stationary phase.

Optogenetics is emerging as an alternative to chemical induction in metabolic engineering[24–26]. Here, light is used as the inducer, and genetically encoded photoreceptor proteins typically regulate transcription of target protein expression[11,12,25–27]. Due to the quantitative and temporal precision with which it can be applied, light can be used to optimize metabolic pathway activity in dimensions that are difficult to access with chemical induction systems. For example, researchers have optimized the timing of light-driven enzyme induction around the exponential to stationary phase transition[28] and the duty cycle of a fixed period light pulse[27] to yield increases in titer of mevalonate in *E. coli* and isobutanol in *S. cerevisiae*, respectively. However, the full capabilities of optogenetic tools have yet to be leveraged for precision

control of metabolic enzyme activity[25,29]. For example, complex induction strategies involving varying light intensity[30] and the period and duty cycle of a light pulse have never been utilized to optimize a metabolic pathway.

The two-component regulatory system CcaSR (Fig. 1a) is among the most widely used bacterial optogenetic tools[31–45]. CcaSR offers several advantages, including photoreversibility, which enables more rapid switching between different target protein expression levels, and non-toxic visible light control wavelengths[30,46]. Here, the sensor histidine kinase CcaS covalently binds the chromophore phycocyanobilin (PCB) to form a holoprotein that is activated by green and deactivated by red light. PCB is synthesized via the oxidation and subsequent reduction of heme via the enzymes heme oxygenase 1 (encoded by *ho1*) and ferredoxin-dependent phycocyanobilin reductase (encoded by *pcyA*). In the active form, holo-CcaS (hereafter CcaS) phosphorylates the response regulator protein CcaR, which binds to the $P_{cpcG2}$ target promoter to activate transcription. Through a series of optimizations, including modulating the expression levels of an engineered *ho1-pcyA* operon as well as *ccaS* and *ccaR*, and truncating poorly-characterized elements from $P_{cpcG2}$, we engineered an exponential phase-optimized system named CcaSR v3.0 that exhibits low leakiness in red light and several hundred-fold activation by green[23,47].

Here, we set out to use CcaSR v3.0 to optimize metabolite production via heterologous enzyme expression in stationary phase. However, we find that this sensor is leaky in red light, resulting in negligible activation by green in these conditions. We discover that PCB biosynthesis is lost in stationary phase due to the metabolic burden of the plasmid-borne *ho1-pcyA* operon. We overcome this issue by relocating *ho1-pcyA* to the chromosome and increasing the expression level to offset the resulting decrease in PCB biosynthesis. Furthermore, we find that CcaS and CcaR accumulate to levels that compromise the light response in stationary phase and re-optimize expression of both proteins to restore proper behavior. We combine the resulting CcaSR$_{stat}$ system with the tyrosine ammonia lyase gene (*tal*) and optimized steady state and dynamic light signals to optimize production of *p*-coumaric acid and demonstrate that these optimizations scale to a 25 mL bioreactor. Finally, we demonstrate that

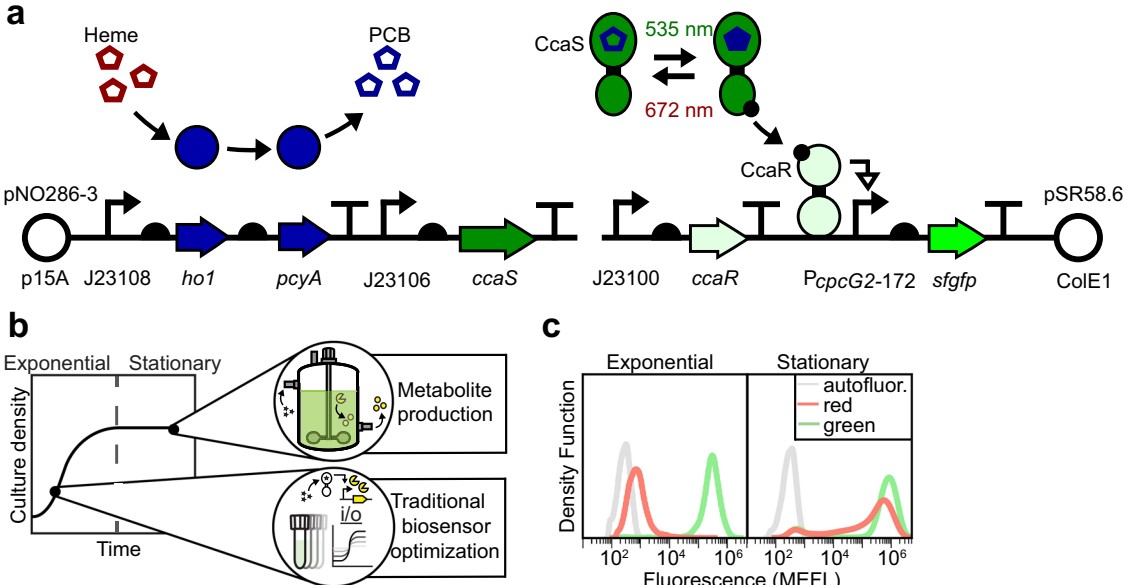

**Fig. 1 | CcaSR v3.0 is not functional in stationary phase. a** CcaSR v3.0 schematic. Semicircles are ribosome binding sites. Vertical lines topped by horizontal lines are transcriptional terminators. **b** Depiction of the difference between exponential and stationary phases wherein bacterial sensors are typically optimized and used, respectively. **c** Fluorescence histograms of the CcaSR v3.0 light response (sfGFP

output) in exponential and stationary phases. Samples were collected at the times denoted in Supplementary Fig. 1b. Non-engineered (autofluorescent) *E. coli* are shown in gray. Data represent one experiment. Data from three biological replicates run on the same day are provided in Supplementary Fig. 1c. Source data are provided as a Source data file.

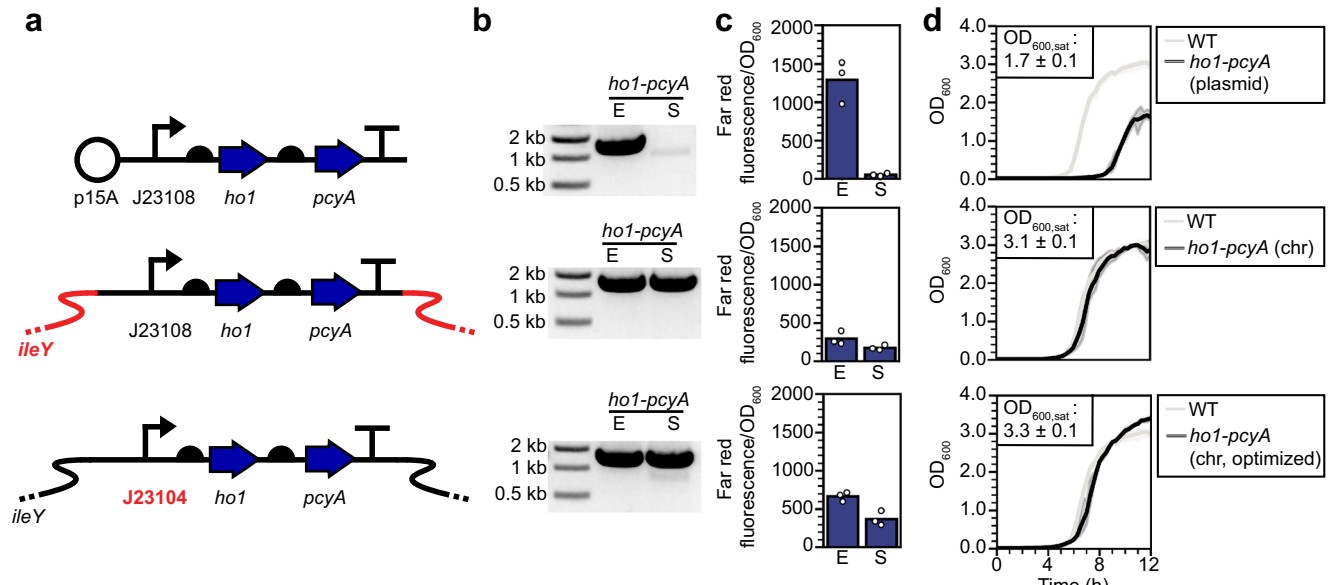

**Fig. 2 | Stabilizing PCB production in stationary phase. a** Schematic of plasmid- (top), initial chromosomal- (center), and optimized chromosomal (bottom) *ho1-pcyA* expression cassettes. Red indicates a genetic modification introduced relative to the cassette above. **b** PCR amplification of the three *ho1-pcyA* cassettes from *E. coli* cultures grown to exponential (E) or stationary (S) phase. Expected size is 1.7 kb. $n = 2$ replicates on 2 separate days were collected. Unmodified gel images and replicate gels are provided in Supplementary Fig. 2. **c** PCB levels produced from each cassette in exponential and stationary phases as reported by Cph1(Y176H):PCB far-red fluorescence. Individual ($n = 3$) replicates on 3 separate days are shown. Bars represent the mean across the three replicates. **d** Growth curves of parent BW29655 (WT) and different PCB producing strains (plasmid or chromosomal (chr), black). $OD_{600, sat}$ indicates the carrying capacity, or maximum $OD_{600}$ reached, by cultures of the engineered strain. Shaded region represents the standard deviation of the mean of $n = 3$ replicate experiments performed on 3 separate days. Source data are provided as a Source data file.

CcaSR$_{stat}$ can be used to optimize the steady-state expression level of larger metabolic pathways, using the betaxanthin family of plant-derived pigments as a model.

## Results

### CcaSR v3.0 is dysfunctional in stationary phase

CcaSR v3.0 is encoded on two multi-copy plasmids (Fig. 1a, Supplementary Fig. 1a). The first (pNO286-3) contains a ~9 copy p15a origin of replication[48], and encodes the J23108 and J23106 constitutive promoters driving expression of *ho1-pcyA* and *ccaS*, respectively. The second (pSR58.6) contains an ~18 copy ColE1 origin[48] and the J23100 promoter driving constitutive *ccaR* expression. Finally, pSR58.6 encodes a superfolder green fluorescent protein (*sfgfp*) reporter gene under control of the optimized CcaR target promoter $P_{cpcG2-172}$.

To characterize the performance of CcaSR v3.0 in stationary phase (Fig. 1b), we initially grew dilute cultures of the *E. coli* K-12 strain BW29655 transformed with pNO286-3 and pSR58.6 aerobically with vigorous shaking at 37 °C in darkness. We chose BW29655 as the host strain because we previously used it for exponential phase characterization of CcaSR v3.0[47]. In these conditions, BW29655 CcaSR v3.0 grows exponentially between 4 and 10 h after dilution and enters stationary phase at approximately 12 h (Supplementary Fig. 1b). While BW29655 cultures lacking the CcaSR v3.0 plasmids reach an optical density at 600 nm ($OD_{600}$) of $2.90 \pm 0.1$, the co-transformed cultures plateau at only $OD_{600} = 1.23 \pm 0.01$ (Supplementary Fig. 1b). Thus, CcaSR v3.0 substantially reduces biomass accumulation of BW29655, likely due to the fact that it consumes cellular heme pools and utilizes protein expression resources.

Next, we grew cultures of BW29655 CcaSR v3.0 as before, but under saturating red or green light (Supplementary Fig. 1c). We sampled these cultures at exponential ($t = 8$ h, $OD_{600} = 0.1$) and stationary ($t = 24$ h, $OD_{600} = 1.2$) phases and used flow cytometry to quantify their respective sfGFP fluorescence levels. In exponential phase, CcaSR v3.0-driven sfGFP fluorescence increases 310-fold from the low level of

$1040 \pm 16$ Molecules of Equivalent Fluorescein (MEFL) in red light (BW29655 autofluorescence = $223 \pm 3$ MEFL) to the high level $330,400 \pm 8700$ MEFL in green (Fig. 1c, Supplementary Fig. 1d), consistent with our previous measurements of CcaSR v3.0 performance[47]. On the other hand, most red light-treated bacteria produce high levels of sfGFP in stationary phase, with green light resulting in little additional induction (Fig. 1c, Supplementary Fig. 1e). At 48 h, these trends remain unchanged (Supplementary Fig. 1f), confirming that behavior at 24 h is reflective of later stationary phase behavior.

### Plasmid-encoded PCB biosynthesis is lost in stationary phase

Our flow cytometry measurements also revealed high cell-to-cell variability in CcaSR v3.0 output in stationary phase. While the single-cell distribution of sfGFP fluorescence exhibits a typical log-normal distribution in exponential phase, it becomes bi-modal in stationary phase, with the higher sfGFP-expressing mode bearing a long tail toward low expression (Fig. 1c, Supplementary Fig. 1e, f). Furthermore, there is a sub-population of low sfGFP-expressing cells that is not light responsive (Supplementary Fig. 1d–f). These effects occur in both defined M9 media and complex LB and 2xYT media (Supplementary Fig. 1g–j), suggesting they are not due to media composition. At high levels, PCB biosynthesis is toxic to *E. coli* and can result in mutations to the *ho1-pcyA* operon[49]. Thus, we hypothesized that these effects could be due to a mutation(s) disabling PCB biosynthesis in stationary phase.

To examine this possibility, we grew dilute cultures of BW29655 transformed with only pNO286-3 (Fig. 2a) to exponential or stationary phase and utilized diagnostic PCR to confirm the presence of the *ho1-pcyA* cassette in both phases (Supplementary Fig. 2). As expected, we observed a robust *ho1-pcyA* cassette PCR product in exponential phase. However, this product was almost completely absent in stationary phase (Fig. 2b, Supplementary Fig. 2c). Whole-plasmid sequencing revealed that the transcriptional terminator encoded at the end of the *ccaS* expression cassette had recombined with a terminator of the same sequence encoded at the end of the *ho1-pcyA*

cassette, resulting in deletion of the latter (Supplementary Fig. 2a, Supplementary Data 1). Such deletions are known to occur in plasmids encoding multiple copies of the same terminator[50]. To confirm that PCB is not produced in stationary phase, we expressed a mutant of the *Synechocystis* PCC6803 phytochrome protein Cph1 (Y176H) that exhibits far-red fluorescence when bound to PCB (Supplementary Fig. 3)[51,52]. We observed strong far-red fluorescence in exponential phase, but very weak far-red fluorescence in stationary (Fig. 2c), consistent with mutation-driven elimination of PCB biosynthesis in the latter. We conclude that the exponential phase-optimized *ho1-pcyA* cassette from CcaSR v3.0 is unstable in stationary phase, leading to a loss in PCB biosynthesis and a compromised light response in most bacterial cells.

### Chromosomal integration of ho1-pcyA stabilizes PCB biosynthesis in stationary phase

To engineer *E. coli* to reliably produce PCB in stationary phase, we first integrated the *ho1-pcyA* cassette from CcaSR v3.0 into the *ileY* locus of the BW29655 chromosome (Methods). Chromosomal integration of toxic heterologous pathways is known to reduce mutation rate and metabolic burden[53,54]. We used diagnostic PCR and DNA sequencing to confirm that this chromosomal *ho1-pcyA* cassette is stable in both exponential and stationary phases (Fig. 2a, b, Supplementary Fig. 2c). Cph1(Y176H) fluorescence confirmed that chromosomal encoding of the exponential phase-optimized *ho1-pcyA* cassette restores stationary phase PCB biosynthesis, albeit to only 13% of the level produced from pNO286-3 in exponential (Fig. 2b, c). To increase stationary phase PCB levels, we replaced the medium-strength J23108 promoter driving *ho1-pcyA* transcription with the stronger J23104 variant (Fig. 2a). This J23104-driven *ho1-pcyA* cassette is also stable in the chromosome (Fig. 2b, Supplementary Figs. 2c and 4) and increases chromosomally-generated PCB levels to 30% of those generated by pNO286-3 in exponential (Fig. 2c). The maximum $OD_{600}$ reached by BW29655 cultures is not reduced by either chromosomal *ho1-pcyA* cassette ($p > 0.05$, paired, 2-tailed *t*-test) (Fig. 2d). Thus, chromosomal integration of *ho1-pcyA* enables stable and non-toxic PCB production in stationary phase. We denote the BW29655 strain carrying the chromosome-optimized (i.e., J23104-driven) *ho1-pcyA* cassette as JTL1.

To examine whether this chromosome-optimized *ho1-pcyA* cassette enables CcaSR function in stationary phase, we co-transformed a variant of pNO286-3 lacking *ho1-pcyA* (pJTL321) with pSR58.6 into JTL1. Although this CcaSR system design produces a 180-fold increase in sfGFP fluorescence in response to green light in exponential phase, this response is reduced to 4.3-fold in stationary phase due to leaky sfGFP expression in red light (Fig. 3b). As CcaSR leakiness and green light response are strongly dependent upon CcaS and CcaR expression level[23], we hypothesized that the poor light response of this strain is due to accumulation of CcaS and CcaR in stationary phase.

### Optimizing CcaS and CcaR expression for stationary phase

To examine CcaS and CcaR levels in stationary phase, we translationally fused *sfgfp* to *ccaS* in the pNO286-3 backbone and to *ccaR* in a modified version of pSR58.6 wherein *sfgfp* is deleted downstream of $P_{cpcG2-172}$. Consistent with our hypothesis, we found that both proteins accumulate to higher abundance in stationary phase (Supplementary Fig. 5). Thus, we moved the *ccaS* expression cassette onto a lower (~3) copy pSC101-origin plasmid[48], resulting in pJTL269 (Supplementary Fig. 5c). We found that this modification reduces stationary phase CcaS levels to those generated by CcaSR v3.0 in exponential phase (Supplementary Fig. 5d). As a result, we performed no further optimizations of *ccaS* expression.

As two-component systems are highly sensitive to response regulator expression levels[55], we constructed a small library of five plasmids wherein the strength of the *ccaR* RBS varies over several orders of

magnitude (Fig. 3c). We used an sfGFP tag to confirm that these RBS variants result in a range of CcaR levels in the cells (Fig. 3c, Supplementary Fig. 6). Next, we individually co-transformed the untagged library plasmids with pJTL269 and a newly-constructed plasmid named pJTL256 wherein the $P_{cpcG2-172}$-*sfgfp* reporter module is encoded on a p15A backbone. We found that the weakest *ccaR* RBS (stationary phase CcaR-sfGFP expression of $2350 \pm 270$ MEFL) results in low $P_{cpcG2-172}$ activity in both red and green light (Fig. 3d). The second weakest RBS (encoded on a plasmid named pJTL257.2, stationary phase CcaR-sfGFP expression of $3150 \pm 250$ MEFL) results in a statistically significant ($p = 3.3 \times 10^{-5}$, Student's two-tailed *t*-test) increase in CcaR-sfGFP expression compared to the weakest (Fig. 3d) as well as the desired properties of low leaky expression in red light and strong activation by green (Fig. 3d). Finally, the three strongest RBSs result in high levels of $P_{cpcG2-172}$ activity in both red and green light, consistent with previous results in exponential phase[23]. Interestingly, the optimal CcaR expression level is the same in stationary and exponential phases (Fig. 3c, Supplementary Fig. 5e, f), suggesting that absolute CcaR level is a critical parameter for light response across growth phases. Due to its strong light response, we named the JTL1 strain containing pJTL256, pJTL269, and pJTL257.2 CcaSR$_{stat}$. Notably, CcaSR$_{stat}$ exhibits no growth defect compared to BW29655 (Fig. 3e), suggesting that the genetic components will be stable and that the strain can accumulate high levels of biomass. Consistent with exponential phase CcaSR systems, CcaSR$_{stat}$ exhibits an intensity-dependent green light response (Fig. 3f). It also exhibits first-order activation kinetics with a time constant ($\tau$) of 16.5 h (Fig. 3g). Though these kinetics are substantially slower than those in exponential phase[47] they are consistent with previous studies of inducible protein expression systems in stationary phase[14] and are likely sufficient for the days-long timescales of fermentations.

During our characterization experiments, we noticed that CcaSR$_{stat}$ does not respond to green light until cultures reach stationary phase (Fig. 3h, Supplementary Fig. 7). We theorized that this specificity is due to insufficient CcaS and/or CcaR levels in exponential phase. Indeed, we constructed a CcaSR$_{stat}$ variant with stronger *ccaS* expression that responds to light in both exponential and stationary phases (Supplementary Fig. 8). The stationary phase specificity of CcaSR$_{stat}$ may be beneficial for metabolic engineering applications, as green light can be applied during growth and production phases, but not result in unwanted expression in the former. Alternatively, the variant that functions in exponential and stationary could enable enzyme induction slightly earlier in the fermentation process, combined with longer-term induction during production phase.

Next, we aimed to understand how CcaSR$_{stat}$ operates in other *E. coli* strains. To this end, we tested CcaSR$_{stat}$ in the human probiotic *E. coli* Nissle 1917 (EcN). As EcN does not share an identical *ileY* locus with K12, we targeted the *torT* and *yhbC* loci for integration of the optimized chromosomal *ho1-pcyA* cassette (Supplementary Fig. 9a). We then transformed the *ccaS* expression plasmid (pJTL269), *ccaR* expression plasmid (pJTL257.2), and *sfgfp* output plasmid (pJTL256) all into both backgrounds to create EcN *torT*:: CcaSR$_{stat}$ and EcN *yhbC*:: CcaSR$_{stat}$, respectively (Supplementary Fig. 9a). Both strains exhibit large green light responses in stationary phase (Supplementary Figs. 9b, d, f) and are inactive in exponential phase (Supplementary Fig. 9c, e). These results show that CcaSR$_{stat}$ can function in diverse *E. coli* strain backgrounds.

### Optimizing phenylpropanoid production with static light signals

To demonstrate the utility of CcaSR$_{stat}$ for metabolic engineering, we next utilized it to optimize production of the phenylpropanoid *p*-coumaric (*p*-CA) acid in *E. coli*. Phenylpropanoids are a large class of mostly plant-derived secondary metabolites produced from L-tyrosine and L-phenylalanine[56]. These compounds have strong antioxidant

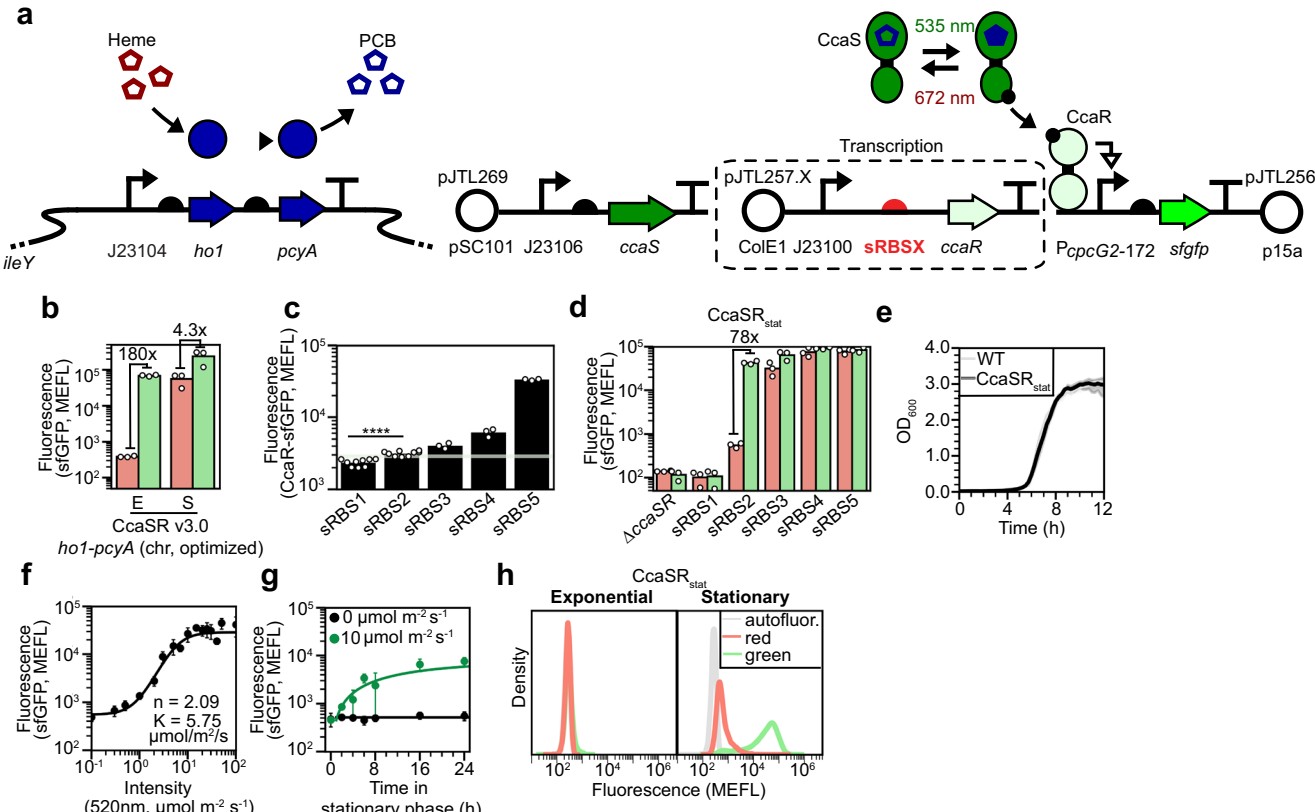

**Fig. 3 | Optimizing *ccaR* expression for stationary phase. a** Optimization of the *ccaR* RBS in JTL1 transformed with the pJTL269 *ccaS* expression and pJTL256 *sfgfp* reporter plasmids. sRBSX indicates one of five synthetic RBSs designed by the RBS calculator (Methods). **b** Exponential (E) and stationary (S) phase light response of CcaSR v3.0 with the optimized chromosomal PCB production system. Bars represent the arithmetic mean of three biologically independent replicates collected on the same day. Numbers indicate the fold change in sfGFP fluorescence between red and green light conditions. **c** Stationary phase CcaR expression levels from each of the five sRBSs. CcaR expression is reported as fluorescence of cultures expressing CcaR-sfGFP fusions under each RBS. Solid horizontal light green shaded region represents the mean of 3 independent biological samples of CcaR-sfGFP expressed from CcaSR v3.0 in exponential phase, collected on the same day. The *p* value between sRBS1 and sRBS2 expression is *p* = 3.3 × 10⁻⁵ (two-tailed Student's *t*-test). **d** sfGFP fluorescence in red and green light for the five sRBS variants. Bars represent the arithmetic mean of three independent biological replicates collected over 2 separate days. CcaSR_stat is based on sRBS2. **e** Growth curves of CcaSR_stat (black) and wild-type BW29655 (gray). Shaded regions represent standard deviation of the mean of three replicates across 3 separate days. **f** Stationary phase green light intensity response and **g** activation dynamics of CcaSR_stat. Light input (either 0 or 10 μmol m⁻² s⁻¹) for activation dynamics characterization supplied at *t* = 0 h. Error bars represent the standard deviation of the mean of three biologically independent replicates collected on the same day. Solid lines represent model fits (Methods). **h** sfGFP fluorescence histograms showing the light response of CcaSR_stat in exponential and stationary phases. Data are representative and represent one experimental trial for each condition. Replicate data over three biologically independent samples collected on the same day are provided in Supplementary Fig. 7e, f. Source data are provided as a Source data file.

properties and applications in the medical and cosmetic industries[57–59]. The production of *p*-CA is the first committed step towards phenylpropanoid metabolism. The most efficient pathway for engineered microbial *p*-CA biosynthesis is expression of a heterologous tyrosine ammonia lyase (*tal*) gene, which encodes an enzyme (TAL) that converts ʟ-tyrosine into *p*-CA in one step (Fig. 4a)[60]. However, TAL-mediated *p*-CA production is toxic in *E. coli* and results in growth defects and lower total fermentation biomass[61]. Due to the stringent and precise control of gene expression it enables, we hypothesized that CcaSR_stat could be used to achieve high *p*-CA titers with minimal toxicity.

To explore this hypothesis, we used CcaSR_stat to express *Streptomyces sp.* NRRL F-4489 *tal*, creating a strain named CcaSR_stat-*tal*. We chose this gene variant because the TAL enzyme it encodes produces *p*-CA with high selectivity[62]. First, we confirmed that we can control TAL abundance with green light by fusing sfGFP to the TAL C-terminus (Fig. 4b) and treating the corresponding cultures with green light intensities between 0.1 μmol m⁻² s⁻¹, which does not activate CcaSR_stat, and 100 μmol m⁻² s⁻¹, which results in full activation (Fig. 3f) for 72 h in 30 °C batch conditions (Fig. 4a). We found that 7 μmol m⁻² s⁻¹ green light results in the highest levels of TAL-sfGFP, and that higher

intensities result in unreliable expression (Fig. 4c), likely due to toxicity. Deletion of CcaSR_stat abolishes light-dependent TAL-sfGFP accumulation (Fig. 4d). We repeated this experiment with a strain expressing untagged *tal* (Fig. 4e) and measured the resulting *p*-CA titers via HPLC. *p*-CA titers increase with green light intensity up to 7 μmol m⁻² s⁻¹, where they reach a high of 71.3 ± 2.8 mg/L (Fig. 4f). Above this intensity, titers become increasingly variable, consistent with toxicity from TAL overexpression (Fig. 4c, f). Bacteria lacking CcaSR_stat produce no detectable *p*-CA in either light condition (Fig. 4g). These data confirm that CcaSR_stat can be used to optimize the expression level of a single-enzyme metabolic pathway in stationary phase.

We next sought to use CcaSR_stat to optimize production of a metabolite produced by a multi-enzyme pathway. Betaxanthins are a class of plant-derived pigments noted for their bright orange color as well as antioxidant and anti-inflammatory properties[63,64]. These compounds are produced via a spontaneous Schiff base condensation between betalamic acid and an amino acid or amine (Supplementary Fig. 10a)[64]. In *E. coli*, betalamic acid can be produced from ʟ-tyrosine in two enzymatic steps via 4-hydroxyphenylacetate 3-monooxygenase (*hpaB*) from *E. coli* W, along with the flavin reductase HpaC, which

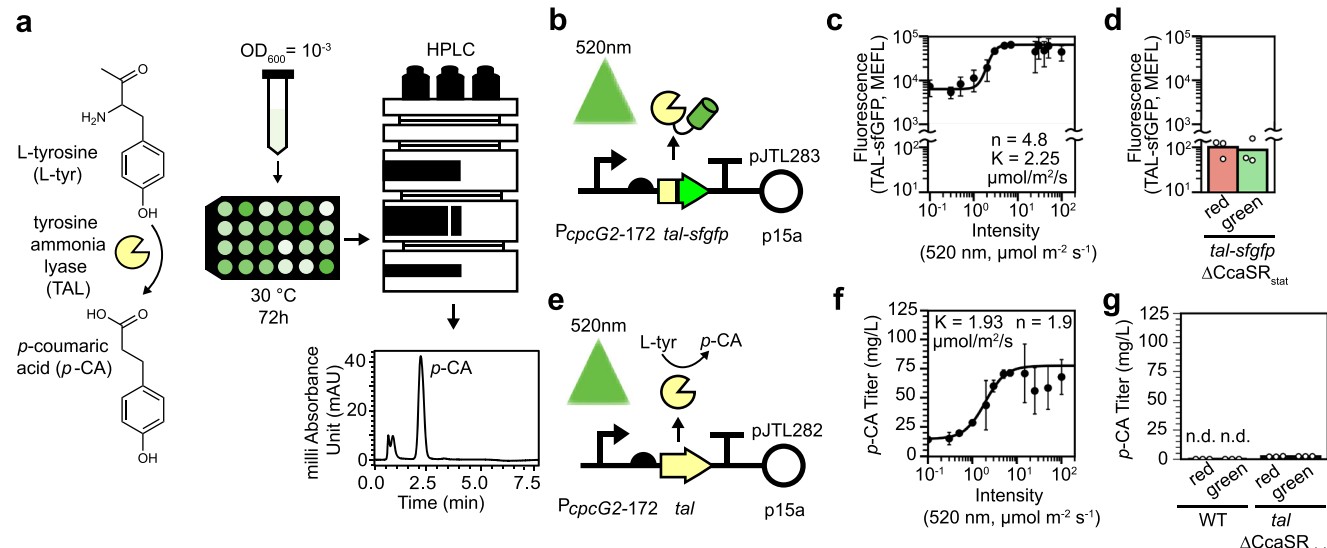

**Fig. 4 | Using static light signals to optimize *p*-coumaric acid production in stationary phase. a** Diagram of the engineered *p*-CA biosynthetic pathway, 24-well plate batch culture fermentation strategy, and HPLC detection method. **b** CcaSR$_{stat}$:*tal-sfgfp* strain, wherein green light induces *tal-sfgfp* expression. **c** Relationship between green light intensity and *tal-sfgfp* expression for CcaSR$_{stat}$:*tal-sfgfp* in stationary phase. Error bars represent the standard deviation of the mean of 3 replicates collected on the same day. **d** Light response of control *tal-sfgfp* strain lacking CcaSR$_{stat}$. Bars represent the arithmetic mean of three biologically independent replicates collected on the same day. **e** CcaSR$_{stat}$:*tal* strain, which is used for *p*-CA production. **f** Relationship between green light intensity and *p*-CA titer. Error bars represent the standard deviation of the mean of thre independent biological replicates collected on the same day. Black lines represent Hill function fits (Methods). K is the green light intensity resulting in half maximal activation of *p*-CA production. *n* represents the Hill coefficient of the fit. **g** Control experiments measuring *p*-CA levels produced by the non-engineered (WT) and *tal* strain lacking CcaSR$_{stat}$. Bars represent the arithmetic mean of three independent biological replicates collected on the same day. Source data are provided as a Source data file.

provides FADH$_2$ to HpaB, and *M. jalapa* 4,5-Dopa dioxygenase (*mjDoda*)[65,66], followed by a spontaneous rearrangement to yield the final product. Thus, we constructed a synthetic operon resulting in co-transcription of these three genes under control of CcaSR$_{stat}$ (Supplementary Fig. 10a). Using LC Q-TOF MS analysis, we identified betalamic acid, dopaxanthin, and portulacaxanthin II as products of our fermentation under green light only (Supplementary Figs. 10b–d and 11–13). Since betaxanthins are intrinsically fluorescent, we used bulk culture fluorescence to assay for betaxanthin levels in higher throughput plate-based assays (Supplementary Fig. 14)[67,68]. Using this method, we found that betaxanthin levels increase with green light intensity up to the maximum of 100 $\mu$mol m$^{-2}$ s$^{-1}$ tested in our experiments (Supplementary Fig. 15). Consistent with the function of CcaSR$_{stat}$, we found that betaxanthins are primarily produced in stationary phase (Supplementary Fig. 16). These experiments demonstrate that CcaSR$_{stat}$ can be combined with static green light signals of variable intensity to optimize the production of metabolites derived from co-transcribed multi-enzyme pathways.

## Dynamic light signals further enhance p-CA production

Pulsatile induction may result in higher product titers than static induction by distributing total enzyme expression across the course of a fermentation while avoiding acute toxicity arising from high intracellular enzyme concentration at any given point in time[26,27]. To explore this possibility in our system, we next grew CcaSR$_{stat}$-*tal* under periodic administration of the optimized green light intensity of 7 $\mu$mol m$^{-2}$ s$^{-1}$ with seven different periods (*T*) varying between 0.02 and 1000 min and eight different duty cycles (dc) varying between 20 and 90% in a constant red light background (Fig. 5a, Supplementary Fig. 17). We exposed the bacteria to all combinations of these periods and duty cycles (56 total conditions replicated on 5 separate days) 6 h after back-dilution of the cultures, an induction time that we found to maximize *p*-CA titer (Fig. 5b, Supplementary Fig. 18). Though there is substantial variability in the data, *p*-CA titers clearly increase with duty cycle (Fig. 5b, Supplementary Fig. 19a). On the other hand, there is

significant effect of light period on *p*-CA titer (Supplementary Fig. 19b). However, we identified two pulsatile light treatment conditions (*T* = 10 min, dc = 70%; *T* = 10 min, dc = 85%) that yield significantly higher *p*-CA titer compared to static 7 $\mu$mol m$^{-2}$ s$^{-1}$ green light (one-tailed Welch's *t*-test *p* = 0.021 and 0.039, respectively). These effects hold for green light intensities between 15 and 100 $\mu$mol m$^{-2}$ s$^{-1}$ but are absent at lesser intensities where *p*-CA titers are much lower (Supplementary Fig. 21). Overall, the best pulsatile light condition (7 $\mu$mol m$^{-2}$ s$^{-1}$, *T* = 10 min, dc = 85%) results in our highest measured titer of 113.1 ± 39.7 mg/L, a substantial increase over the maximum titer we achieved with constant light treatment (Fig. 4f).

The optimized static light signal results in lower CcaSR$_{stat}$-*tal* culture densities than the optimized dynamic light signal (Supplementary Fig. 20a). This effect does not occur in non-engineered *E. coli* (Supplementary Fig. 20a), indicating that it arises due to TAL expression differences. However, using CcaSR$_{stat}$-*tal-sfgfp*, we find that TAL expression levels are identical in static and pulsatile light conditions over the large majority of a 3-day Light Plate Apparatus (LPA) fermentation (Supplementary Fig. 20b). These data indicate that dynamic TAL induction is less toxic to *E. coli* than static TAL induction at the high levels needed for optimal *p*-CA titers. Pulsatile light may reduce the toxicity of TAL expression by periodically allowing cells to regenerate L-tyrosine pools or liberating RNA polymerases and ribosomes to produce housekeeping or other gene products important for bacterial physiology. Ultimately, the reduced toxicity of pulsatile TAL induction results in more biomass for *p*-CA production.

Next, we applied similar period/duty cycle optimizations at the optimal static light intensity of 100 $\mu$mol m$^{-2}$ s$^{-1}$ (Supplementary Fig. 15) to further optimize our engineered betaxanthin pathway. However, pulse optimization failed to increase betaxanthins beyond levels achieved with static light treatment at the same intensity (Supplementary Fig. 22). These data suggest that multi-step biosynthetic pathways may require independent optimization of the induction dynamics of different enzymes.

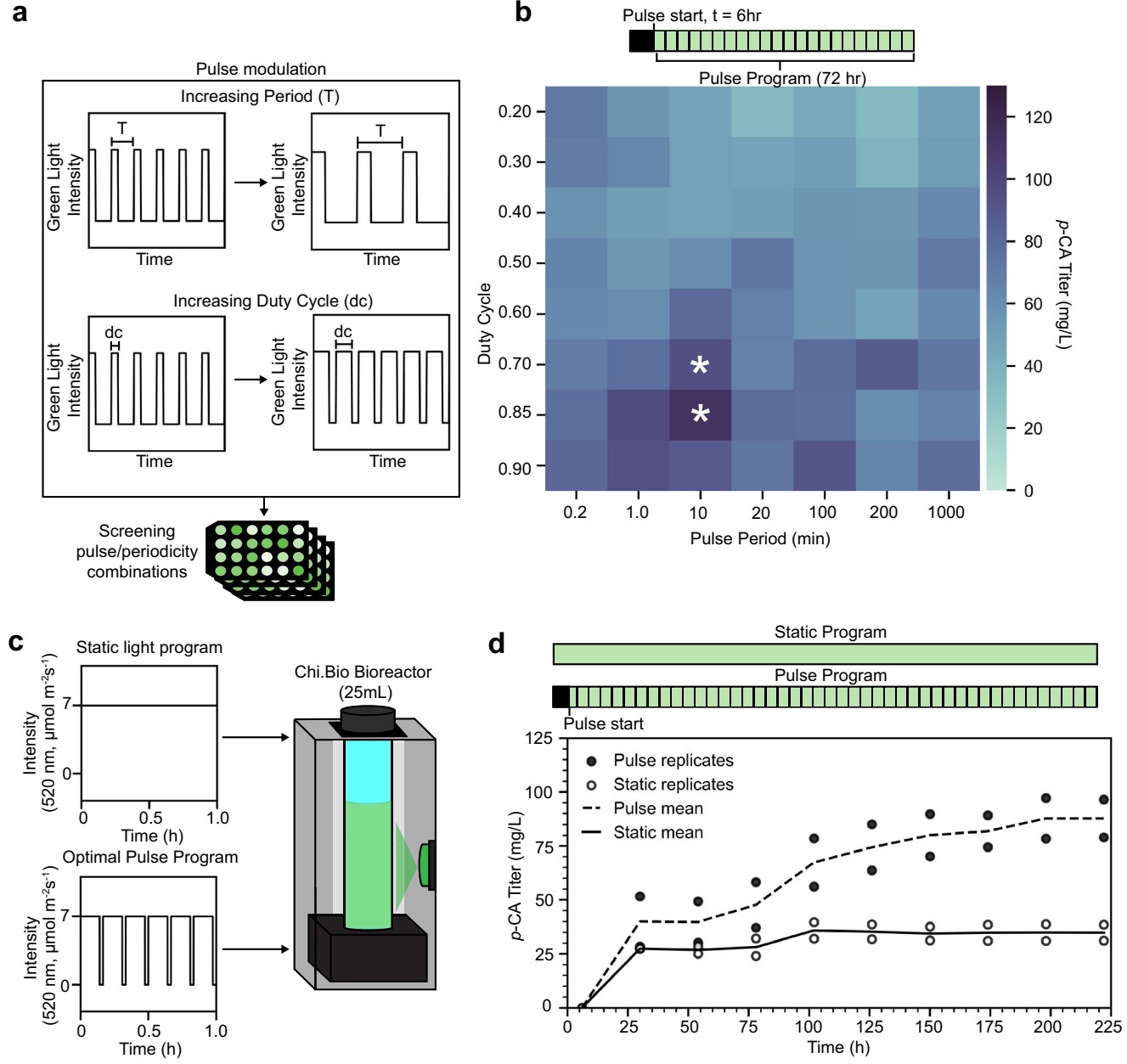

**Fig. 5 | Using dynamic light signals to optimize *p*-CA production in stationary phase. a** 24-well plate experiments modulating the period (*T*) and cycle (dc) of green light applied to the CcaSR$_{stat}$:*tal* strain. **b** Relationship between *T*, dc, and *p*-CA titer. Periodic green light treatment was initiated 6 h after the culture was back-diluted and lasted 72 h. Asterisks indicate statistically significant increases (one-tailed Welch's *t*-test, *p* < 0.05) compared to the maximal *p*-CA titers achieved with static light intensity optimization (Fig. 4f). Each colored square represents the mean of five biologically independent replicates collected on 5 separate days. **c** Optimized static and dynamic light programs were tested in a larger volume bioreactor with a programmable green light source. **d** *p*-CA titers over time for the CcaSR$_{stat}$:*tal* strain treated with the optimized static and dynamic (pulse) light programs. Titers of *n* = 2 replicates collected over the same time period are plotted. Source data are provided as a Source data file.

## Dynamic light induction enhances p-CA titer in a bioreactor

We next examined how our batch fermentation results translate to a bioreactor format. To this end, we compared our optimized static and dynamic green light signals, which were conducted at 0.5 mL culture volume (Methods), to 25 mL commercial reactors capable of exposing growing cultures to programmed light signals (Methods)[69] (Fig. 5c). We first developed a methodology based on live bacteria and *p*-CA measurements to calibrate the reactors to emit an intensity of green light intensity operationally equivalent to the optimized 7 μmol m$^{-2}$ s$^{-1}$ level from our 0.5 mL system ("Methods," Supplementary Fig. 23). Next, we measured the effect of the optimal static and periodic light signals on *p*-CA titer in these reactors. For the static

signal, we found that *p*-CA titers increase for 4 days and then saturate at 34.8 mg/L (Fig. 5d). Strikingly, *p*-CA titers increase for 8 days in response to the dynamic signal and reach a higher final titer of 87.8 mg/L (Fig. 5d). The dynamic signal results in faster bacterial growth and higher culture carrying capacity (OD$_{600}$ = 1.5 versus 1.0) than the static signal (Supplementary Fig. 24), which explains the higher titer, at least in part. Carrying capacity decreases from OD$_{600}$ = 2.1 in our batch experiments (Supplementary Fig. 20a) to OD$_{600}$ = 1.4 in our reactor conditions (Supplementary Fig. 24), which may explain the lower *p*-CA titer in the latter. These data demonstrate that metabolic pathway optimizations using CcaSR$_{stat}$ can scale from batch to bioreactor settings. Future reactor optimizations to feed

strategies and strain mixing will allow for increases in cell densities, which will in turn result in higher *p*-CA titer.

## Discussion

As synthetic biology increasingly moves toward real-world applications, researchers must focus on designing genetic systems that function outside of standard laboratory conditions[70,71]. To this point, there has been little focus on understanding how synthetic genetic systems optimized to function in exponential phase perform in stationary phase[14,72]. Here, we begin to address this need by characterizing how the performance of the well-characterized CcaSR gene regulatory system changes between exponential and stationary phases. Our work shows that there can be dramatic reductions in performance between these two conditions and highlights mutation of metabolically burdensome pathways and accumulation of regulatory proteins as major challenges, in agreement with previous work[9,22,49,73–75]. Encouragingly, we find that established methods of reducing genetic burden and tuning gene expression level are sufficient to overcome these obstacles.

CcaSR$_{stat}$ has several design features that make it well-suited to metabolic engineering. First, its low leakiness and stationary phase dependence prevent unwanted competition for cellular resources prior to induction of enzyme expression. These features are reminiscent of density-dependent quorum-sensing systems, which engineers have repurposed to achieve strong induction of gene expression in stationary phase[76]. However, unlike quorum-sensing systems, which induce target enzymes in a digital-like fashion, the light-intensity dependence of CcaSR$_{stat}$ enables fine-tuning of enzyme expression level. Finally, CcaSR$_{stat}$ is functional in both laboratory-adapted and probiotic *E. coli*, suggesting that it can function in diverse strains, including those commonly used for metabolic engineering.

CcaSR$_{stat}$ could be further improved in future work. Most notably, the *ccaS*, *ccaR*, and P$_{cpcG2-172}$ cassettes could be moved from their respective plasmids into the chromosome. The expression levels of *ccaS* and *ccaR* could then be tuned again to restore the plasmid-derived expression levels established here. Furthermore, the RBS of the target gene(s) could be optimized to increase expression despite chromosomal encoding of P$_{cpcG2-172}$. That said, it may be desirable to encode P$_{cpcG2-172}$ on a plasmid to simplify the swapping of target genes while achieving higher maximum target gene expression levels. The precision with which target genes can be expressed from CcaSR$_{stat}$ could also be improved by knocking out *rmf*, *relA*, and *spoT*, which are known to increase expression noise in stationary phase[77].

The advantages of optogenetic control for metabolic engineering have been discussed previously[25,26]. However, most studies have taken advantage of only a subset of these benefits. Here, we vary the intensity of the inducing wavelength, the timing of light induction, the period and duty cycle of an activating light pulse, and apply inhibitory red light to more quickly deactivate expression in dynamic induction experiments[30,46]. These conditions would be more difficult or not possible to implement using chemical inducer systems, highlighting the benefits of light control. The features of low leakiness, high dynamic range, photoreversibility, stationary phase specificity, and low metabolic burden may provide CcaSR$_{stat}$ advantages over other optogenetic tools that have been used in metabolic engineering[12,25]. Indeed, when applied to optogenetic control of TAL expression, our approach resulted in *p*-CA titers consistent with the highest reported values for a non-tyrosine-overproducing *E.coli* strain[60].

Translating bench-scale optimizations into larger reactor environments is a central challenge in metabolic engineering. Our optimized pulsatile light signal yields higher bacterial densities and *p*-CA titers than static light treatment at both 3-day 0.5 mL batch and 9-day 25 mL fed-batch reactor scales. Interestingly, the pulsatile signal results in continuous *p*-CA production for 8 days, while production ceases

after 4 days when cells are treated with the static light signal in our reactor conditions. This effect is likely due to reduced toxicity arising from periodic compared to steady-state enzyme induction. At larger reactor volumes, poor light penetration is expected to become a major challenge[26,27]. While photobioreactors designed for enhanced light penetration could address this challenge, we believe that optogenetics is currently best suited for the identification of gene expression patterns that increase fermentation productivity. After light has been used to identify optimal induction patterns for a given pathway, it could be replaced with gene[78] or protein circuits (e.g., oscillators)[79] that recapitulate those enzyme expression patterns. In this case, no inducer would be required to achieve optimal titers at industrially relevant fermentation volumes.

In contrast to the single-enzyme *p*-CA pathway, pulsatile activation of the multi-enzyme betaxanthin production pathway failed to significantly improve titer over static green light illumination. One possible explanation is that multi-enzyme pathways may require independent optimization of the expression dynamics of each enzyme. For example, sequential activation of pathway enzymes may improve overall flux from substrate to product[15,41]. To this end, we and others have previously co-expressed CcaSR with blue- and red-light responsive two-component systems and applied multiple light wavelengths to achieve independent control of the transcriptional output of each system[31,41,46]. Future work to optimize such orthogonal light sensors for stationary phase could allow optogenetics to be used to optimize the expression dynamics of multiple pathway enzymes independently. For example, the red-light-activated sensor Cph8-OmpR-cI relies upon the PCB chromophore and could be expressed alongside CcaSR$_{stat}$ in JTL1 or another strain bearing the optimized chromosomal *ho1-pcyA* expression module. The *ho1-pcyA* promoter may need to be further strengthened to accommodate the PCB requirements of both CcaS and the engineered red-light sensor kinase Cph8. Expression of Cph8, the response regulator OmpR, and the cI transcriptional repressor, which inverts red light deactivation to red light activation, could be optimized via expression from different plasmid copy numbers and using RBS libraries, following the approaches we take here. A similar approach could be taken for the engineered blue-light-activated sensor based on the photoreceptor kinase YF1, the response regulator FixJ, and the transcriptional repressor SrpR[41]. This system utilizes flavin mononucleotide, which is naturally produced by *E. coli*, as a chromophore, thus preventing additional competition for PCB. Other blue and red-shifted bacterial optogenetic tools could likely be optimized for stationary phase and used alongside CcaSR$_{stat}$ as well[79–81].

Other synthetic biology applications require reliable sensing in stationary phase and could benefit from the work we perform here. For example, earlier versions of CcaSR have been used to control production of a host longevity-enhancing polysaccharide and a Pb-detoxifying protein from *E. coli* residing within the gastrointestinal tracts of live *C. elegans* worms and *D. melanogaster* fruit flies, respectively[40,45]. In general, bacteria are expected to grow slowly or be in stationary phase in the gut. Indeed, the green light response of CcaSR was shown to be far lower in *C. elegans* than in exponential phase in vitro[45]. CcaSR$_{stat}$ could exhibit substantially larger light responses in such in vivo systems, possibly allowing for longer-term control of gut bacterial function. Chemically-responsive two-component systems have also been used for early medical and environmental applications[55], including detection of gut biomarkers[82], virulence-associated antimicrobial peptides[83], and soil nitrate levels[84]. Such applications would also benefit from sensors optimized for stability and stimulus responsiveness in stationary phase. Thus, the stationary phase sensor design strategy outlined in this work should be applicable to other two-component systems, enabling robust, long-term stimulus detection for a wide range of synthetic biology applications.

## Methods

### DNA cloning

The *ccaS*, *ccaR*, and *ho1-pcy*A cassettes, P*cpc*G2-172, and the *sfgfp* gene were amplified from previous *E. coli* plasmids[47]. The *hpaB* and *hpaC* genes were synthesized by Twist Biosciences based on the *E. coli* W sequence[85]. The *M. jalapa mjDoda* gene was synthesized by Twist using the sequence codon-optimized for *E. coli* by Hou et al.[65]. The *cph1*(Y176H) gene was amplified from a *B. subtilis* PCB biosensor integration module[34]. *Streptomyces* sp. NRRL F-4489 *tal* was codon-optimized for *E. coli* and synthesized (Twist Biosciences). The RBS BCD7, which functions relatively consistently independent of genetic context[86], was constructed via oligonucleotide annealing and cloning into pJTL256. The RBS BCD25k was amplified from a previous *E. coli* plasmid[87]. The RBS calculator (version 2020)[88] was used to predict *ccaR* RBS strengths (Supplementary Fig. 6b). Golden Gate assembly[89] was utilized for all plasmid construction. Clonetegration[54] was utilized for chromosomal integration of the *ho1-pcyA* cassettes. We confirmed integration into the *ileY* locus through PCR amplification with primers binding directly outside of *ileY* and performing Sanger sequencing on the PCR product.

### Strain preparation

All plasmids were chemically transformed into their respective background strain (Supplementary Data 2). Experimental strains were streaked from -80°C 25% v/v glycerol stocks onto Petri dishes with Luria Broth (LB) + 1.2% w/v agar. A single colony was isolated from the plate, transferred to 3 mL of M9 media (see Media Preparation) with appropriate antibiotics in a 14 mL culture tube, and incubated in a shaking incubator at 37 °C and 250 RPM. Once the cultures reached $OD_{600}$ ~ 0.05–0.3, tubes were removed from the incubator. 1400 μL of culture was mixed with 600 μL of 50% v/v glycerol. $OD_{600}$ of the culture-glycerol was then measured; following, 0.1 mL of the mixture was aliquoted into 0.2 mL PCR tubes and stored at −80 °C. These aliquots were used for downstream experiments.

### Media preparation

All experiments utilized defined minimal M9 media. The defined M9 media consisted of 5x M9 salts (Teknova M1902), 2 mM $MgSO_4$, 0.1 mM $CaCl_2$, and 0.2% casamino acids. For phenylpropanoid and betaxanthin fermentation experiments, 200 mg/L L-tyrosine was added to the media along with 5 g/L glucose. No further L-tyrosine could be added to the media without resulting in precipitation. For all other experiments, the media was supplemented with only 5 g/L glucose. All M9 media was filtered through a 0.2 μm filter and stored at room temperature. At the start of an experiment, appropriate antibiotics (100 μg/mL kanamycin, 50 μg/mL spectinomycin, 50 μg/mL ampicillin, and 34 μg/mL chloramphenicol) were added along with 1 mM 5-aminolevulinic acid (Thermo 103925000), a precursor for phycocyanobilin (PCB) biosynthesis. For betaxanthin fermentations, L-ascorbic acid was added at a final concentration of 10 mM as an antioxidant to prevent spontaneous oxidation of L-DOPA[51].

### Non-bioreactor experiments

All non-bioreactor 24-well optogenetic experiments were performed in four LPAs[90]. Each LPA was outfitted with a green 520 nm LED and a red 660 nm LED in each well. LPAs were mounted directly to a shaking incubator. LED calibration was performed as previously described[34,90]. Briefly, a 15.2 cm diameter integrating sphere (StellarNet IS6) attached to a spectrometer (StellarNet UVN-SR-25 LT16) was used to measure photon flux of each LED with a constant programmed intensity, and this flux was used to calculate adjusted dot correction values for each LPA in order to achieve maximal intensities of 100 μmol $m^{-2}$ $s^{-1}$ for both green and red LEDs. For basic static and dynamic light experiments, a custom Python script was used to randomize sample locations across the plate and calculate LPA grayscale values (GSint), resulting in

desired light intensities. GSint values were then input into IRIS v1.0.0[90] and the program.lpf files for each LPA were downloaded. For the pulse function optimization, a custom Python script was created to generate the pulse patterns and save the calibrated GSint patterns into a program.lpf file to be used by the LPA.

On the day of an experiment, frozen 0.1 mL glycerol aliquots were removed from a −80 °C freezer and thawed at room temperature. For non-metabolite production experiments, aliquots were diluted to $OD600 = 10^{-5}$ in the appropriate M9 media. For phenylpropanoid fermentations, we found that a starting $OD_{600} \geq 10^{-3}$ results in slightly higher *p*-CA production (Supplementary Fig. 18a). Thus, all fermentations began at an $OD_{600}$ of $10^{-3}$. Media was then aliquoted into 24-well plates with 500 μL culture per well. For experiments not involving light treatment, clear 24-well plates (Fisher Scientific, 07-200-84) were utilized. For optogenetics experiments, 24-well dark-walled, clear-bottomed plates (ArcticWhite AWLS303008) were used. Plates loaded with inoculated media were sealed with adhesive foil (VWR 60941-126) and mounted onto LPAs in the shaking incubator at 250 RPM. Unless otherwise specified, all wells were supplied with 10 μmol $m^{-2}$ $s^{-1}$ red light in addition to the specified green light signal. Saturating green light was set to 10 μmol $m^{-2}$ $s^{-1}$ for Figs. 1c, 3b, d, h and 4d, g, Supplementary Figs. 1c–j, 7b–f, 8b, c and 9b–f and 100 μmol $m^{-2}$ $s^{-1}$ for Supplementary Figs. 10b–d and 11–13. In other optogenetic experiments, light intensities are indicated in the figure or figure caption.

For exponential phase characterization experiments, bacteria were grown for 8 h at 37 °C to $OD_{600} = 0.05–0.5$. For stationary phase characterization experiments, bacteria were grown until growth stagnated (9–14 h depending on the strain) followed by an additional 12 h in stationary phase (21–26 h total) at 37 °C. For fermentations and TAL-sfGFP expression characterization experiments, bacteria were grown at 30 °C for 72 h (*p*-CA) or 24 h (betaxanthins) in the noted light conditions. After the experiments were complete, the plates were removed from the incubator and immediately placed on an ice slurry for appropriate processing and analysis.

### Plate reader absorbance and fluorescence detection

To assay growth dynamics, 0.1 mL glycerol aliquots were thawed at room temperature and diluted to $OD_{600} = 10^{-5}$ in M9 media. 200 μL of inoculated medium was then pipetted into one well in a black-walled, clear-bottomed 96-well plate (VWR 82050-748) and sealed with adhesive foil. The plate was then inserted into a Biotek Synergy H1 Microplate Reader. Utilizing the associated Biotek software (Gen5 v3.08), the plate reader was programmed to shake for 12–24 h at 37 °C, stopping every 15 min. to quantify absorbance at 600 nm ($A_{600}$). The $A_{600}$ value of a media-only sample was then subtracted from the resulting absorbance data and converted to $OD_{600}$ utilizing the linear relationship:

$$OD_{600} = 2.2 \times A_{600, \text{plate reader, media subtracted}}$$

For Cph1(Y176H) and betaxanthin fluorescence measurements, cell cultures were diluted 1:10 in PBS, and 200 μL of each sample was pipetted into black, clear-bottomed 96-well plates. A program was created to measure both $A_{600}$ and fluorescence signal arising from excitation at 644 nm and emission at 672 nm (the peak excitation/emission of fluorescent Cph1(Y176H)[51]) or excitation of 485 nm and an emission of 515 nm (for detection of betaxanthin pigments[67,68]). For each sample, fluorescence was divided by $OD_{600}$ to reflect per-cell values.

### PCR and sequencing of ho1-pcyA expression cassettes

To analyze *ho1-pcyA* expression cassettes, 50 μL of culture was transferred into a 0.2 mL microtube and boiled at 98 °C for 10 min. Boiled cultures were added to a PCR mixture containing primers PJL01 and PJL02 (Supplementary Data 1) to amplify the expression cassette. For

plasmid-based *ho1-pcyA* strains, the remaining 450 μL of culture was plasmid-purified and submitted to Plasmidsaurus Inc. for whole-plasmid sequencing.

## Bioreactor experiments

Four 25 mL reactors were purchased from Chi.Bio (Berlin, Germany) and assembled according to the manufacturer's instructions. Custom Python programs were written to modulate light levels dynamically and implemented into the ChiBio OS app.py (ChiBio Operating System v1.0). To calibrate the optimal 7 μmol m$^{-2}$ s$^{-1}$ of green light to the arbitrary intensity units of the Chi.Bio reactor, we measured TAL-sfGFP levels in the LPA at 7 μmol m$^{-2}$ s$^{-1}$ of green light after a 72 h, 30 °C fermentation (Supplementary Fig. 23b). We then grew CcaSR$_{stat}$ with the TAL-sfGFP output in the Chi.Bio reactor with varying light conditions, matching the LPA fermentation conditions (72 h, 30 °C) to determine the arbitrary unit green light intensity that results in similar TAL-sfGFP levels to match optimal LPA values. Fortuitously, the lowest testable intensity in the Chi.Bio reactor (0.003 au) results in a value of TAL-GFP of 59,730 ± 1575, which closely matches optimal TAL-GFP expression in the LPAs (65,500 ± 2800 MEFL). Higher intensities of green light in the Chi.Bio reactor resulted in lower TAL expression (Supplementary Fig. 23c), likely due to both phototoxicity and TAL toxicity, a phenomenon we also observed in LPAs at high intensities.

To begin the experiment, 0.1 mL glycerol aliquots of the specified strains were thawed at room temperature. To generate enough cells for the seed culture, the aliquots were diluted 1:100 in 3 mL of M9 media supplemented with 200 mg/L L-tyrosine and grown for 3–4 h until an OD$_{600}$ = 0.05–0.2 was reached. This culture was then diluted to OD$_{600}$ = 10$^{-3}$ in 15 mL M9 media with 200 mg/L L-tyrosine and transferred to a 25 mL glass vial (Fisher 14-955-320) outfitted with a Chi.Bio-specific opaque lid. Vials were then fastened inside the reactors, which were set to an internal temperature of 30 °C with a stirring rate of 0.7, and each optogenetic program was started. Starting at $t$ = 6 h (start of the pulse induction), reactors were sampled every 24 h. To do so, programs were briefly paused, and 500 μL of media from each reactor chamber was removed. At the first five timepoints, a bolus 1 mL M9 with 200 g/L glucose and 200 mg/L L-tyrosine was also added to offset total volume loss from sampling.

## Flow cytometry measurements of sfGFP fluorescence

Culture samples were diluted to an OD$_{600}$ ~ 0.01 in ice-cold PBS supplemented with 1 mg/mL of chloramphenicol to halt translation. Diluted samples were then transferred to a 37 °C water bath and incubated for 15 min to allow sfGFP maturation[30]. Samples were then removed and placed back in an ice slurry for flow cytometry analysis. Cytometry was performed with a Beckman Coulter Cytoflex S using a 488 nm excitation laser and a 525/40 nm emission filter. 20,000 events per sample were collected. In addition, 10,000 events of a suspension of calibration beads (Spherotech RCP-30-5A) were collected at the end of flow cytometry data collection.

The resulting.fcs files were analyzed utilizing the FlowCal software package v1.3[91], following the processing workflow in Supplementary Fig. 25. Samples were density-gated by forward scatter/side scatter, retaining 50% of the total events and removing unwanted debris[91]. Calibration bead samples were similarly density-gated to retain solely 30% of the total events[91]. We then used bead data to convert all sfGFP data from arbitrary units into MEFL values as previously described[90]. Specifically, *E. coli* autofluorescence was subtracted from total sample fluorescence utilizing the following equation:

$$\text{MEFL}_{\text{Sample, AF subtracted}} = \text{MEFL}_{\text{Sample, raw}} - \mu_{AF}$$

where MEFL$_{\text{Sample, AF subtracted}}$ represents the reported, background-subtracted MEFL value of the sample, MEFL$_{\text{Sample, raw}}$ is the raw MEFL value of the sample data, and $\mu_{AF}$ is the arithmetic mean of the MEFL value of $n$ = 3 biological replicates of the autofluorescence control. For all CcaSR v3.0-sfGFP samples, BW29655 was used as the autofluorescence control. For all other samples, the control was JTL1. Autofluorescence values are not subtracted from any plotted histograms. Rather, histograms of autofluorescent bacteria are shown alongside those of engineered fluorescent bacteria for reference. Histograms are represented with kernel smoothing, and raw bin data can be found in the source data file.

## HPLC detection of phenylpropanoids

For HPLC analysis of *p*-CA, culture samples were prepared in 1.5 mL conical tubes and centrifuged for 2 min at 10,000 × *g*. Supernatant was then filtered through a 0.2 μm polyethersulfone (PES) filter (VWR, 76479-024) and stored at −20 °C before HPLC analysis. Filtered samples were then diluted 1:10 in 100% methanol and pipetted into glass chromatography vials. Liquid chromatography was then performed with a Shimadzu HPLC column oven (Shimadzu CTO-20A) with a C18 column (5 μM, 50 × 3 mm) equipped with an autosampler (Shimadzu SIL-20ACHT), quaternary solvent delivery unit (Shimadzu LC-20AD), and UV-Vis detector (Shimadzu SPD-M20A). To detect *p*-CA, an isocratic method was developed based on past literature[92]. The protocol utilized 65% solvent A (1.3% v/v glacial acetic acid) and 35% solvent B (100% methanol) with a rinse solution of 100% methanol. 10 μL of sample was injected and run for 12 min isocratically with a flow rate of 0.2 mL min$^{-1}$ at 28 °C. A *p*-CA standard (VWR IC10257610) was used to create a calibration curve. LabSolutions v.5.9.7 software was utilized to analyze HPLC data. For every sample, the corresponding peak at 306 nm with a retention time of ~6 min. was utilized to calculate *p*-CA peak area. A *p*-CA calibration curve was then utilized to calculate the sample titer.

## LC Q-TOF MS detection of betaxanthins

Culture samples were centrifuged for 5 min. at 5000 × *g* and the supernatant was transferred to a clean, pre-labeled microcentrifuge tube and stored at −20 °C. Betalamic acid, dopaxanthin, and portulacaxanthin II were then identified and qualitatively analyzed using an Agilent 6545XT Q-TOF mass spectrometer (Agilent, US) equipped with a Dual AJS ESI source, operating in positive ion mode and scanning a mass range of *m/z* 100–1200. Chromatographic separation was carried out using an Agilent 1290 Infinity II binary pump at a flow rate of 0.400 mL/min. The LC gradient started at 98% solvent A (0.1% formic acid in water) and 2% solvent B (0.1% formic acid in acetonitrile) from 0 to 2 min, ramped linearly to 50% B between 2 and 12 min, then increased to 98% B from 12 to 15 min, followed by a 2-min hold at 98% B. A post-time of 3 min was applied for re-equilibration. The column compartment was maintained at 35 °C, and the autosampler was set for an injection volume of 10.0 μL. For mass spectrometry, a capillary voltage of 4 kV with a nozzle voltage of 500 V was used. The drying gas flow rate was set to 11 L/min at a temperature of 325 °C, and the sheath gas flow rate was set to 12 L/min at a temperature of 275 °C. Traces were collected and analyzed using an Agilent MassHunter Workstation v10.1 for LC/Q-TOF and MassHunter Qualitative Analysis software v10.0, respectively. The engineered bacteria exposed to green light displayed new chromatogram peaks at retention times of 4.83, 5.08, and 5.354 min (Supplementary Figs. 11–13) that are absent from control samples, including engineered bacteria exposed to red light alone. The corresponding MS spectra indicate *m/z* ratios of 391.1132, 212.0548 and 375.1192 ([M + H]$^+$) suggesting the peaks belong to dopaxanthin (expected ([M + H]$^+$) *m/z* = 391.1136), betalamic acid (expected ([M + H]$^+$) *m/z* = 212.0553), and portulacaxanthin II (expected ([M + H]$^+$) *m/z* = 375.1187). The difference between the experimental *m/z* and expected *m/z* is 1.08 ppm, 2.3 ppm, and 1.27 ppm, respectively, which are all within the instrument's margin of error.

## Model fitting

Green light intensity transfer functions were fit to the Hill function:

$$y = y_{\min} + (y_{\max} - y_{\min}) * \frac{x^n}{x^n + K_{1/2}^n}$$

where $y$ is sfGFP fluorescence, TAL-sfGFP fluorescence, or $p$-CA titer. $y_{\min}$ and $y_{\max}$ represent the minimum and maximum $y$ values in response to no green light and saturating green light, respectively. $x$ represents the green light level in units of μmol m$^{-2}$ s$^{-1}$. $K_{1/2}$ is the intensity resulting in half-activation. $n$ is the Hill coefficient.

For modeling activation kinetics, we utilized a first-order transfer function with delay, following the format:

$$y = \begin{cases} y_{\min} & \text{if } t \leq 0 \\ K * \left(1 - e^{\frac{-t}{\tau}}\right) & \text{if } t > 0 \end{cases}$$

Here, $K$ indicates the gain, or difference between the on and off state of the system, $t$ is time elapsed since $t = 0$ h or the start of green light induction in stationary phase, and $\tau$ is the time constant. To aid in the fitting of our activation kinetics, we specified the gain of the system based on the saturating stationary phase activation signal after 24 h ($7660 \pm 1510$ MEFL). All Hill functions and first-order transfer functions were fit to their respective data sets utilizing the LmFit 1.0.1 python package[93], which utilizes the Levenberg-Marquardt algorithm to perform non-linear least-squares minimization. Pearson's correlation coefficient $R$ and $R^2$ were calculated utilizing Scipy 1.7.3[94].

To generate Bode diagrams, we utilized the bode function of Scipy 1.7.3[94]. Inputting the fitted parameters of the first-order transfer function for CcaSR stationary phase activation, we generated the corresponding Bode reported magnitude and phase data, which were then plotted utilizing Matplotlib 3.5.3 on Python.

## Statistics and reproducibility

To determine statistical significance between strains/conditions/etc, one/two-tailed Student's $t$-tests were deployed. Information on the $p$-values of each $t$-test is clearly stated in the manuscript. Chi.Bio reactor experiments were performed in duplicate due to long time scales and low throughput of experimentation. All other experiments contain at least $n = 3$ biological replicates. In the case of light intensity/pulsing optimizations, the light conditions were randomized, and the scientist performing the experiment was blinded to the ordering of the light conditions per plate.

## Reporting summary

Further information on research design is available in the Nature Portfolio Reporting Summary linked to this article.

## Data availability

The raw flow cytometry data (.fcs) files for Figs. 1c, 3b–d, 3f–h and 4c, d, Supplementary Figs. 1c–j, 3d, 5b, d, f, 6b, 7c–f, 8b, c, 9b–f, 20b and 23b, c, raw HPLC data (.lcd) files for Figs. 4f, g and 5b, d, Supplementary Figs. 18d and 21b, raw LC-MS data (.d) files for Supplementary Figs. 10b–d and 11–14, raw gel images for Fig. 2b, Supplementary Figs. 2c and 4b, and raw sequencing data for Supplementary Fig. 2 have been posted to Figshare (https://doi.org/10.6084/m9.figshare.30203842)[95]. *ho1-pcyA* specific primers and plasmids, strains, plasmids, and genetic parts used in this study are listed in Supplementary Data 1–4, respectively. CcaSR_stat *ccaS* (pJTL269), *ccaR* (pJTL257.2), P$_{cpcG2-172}$:*sfgfp* (pJTL256), P$_{cpcG2-172}$:*tal* (pJTL282), P$_{cpcG2-172}$:*tal-sfgfp* (pJTL283), and P$_{cpcG2-172}$:Betaxanthin pathway (pDJH019) plasmids, and the *cph1*(Y176H) expression plasmid (pSC0025) are available from Addgene with Accession IDs given in Supplementary

Data 3. All strains and other plasmids used in this study are available under a Materials Transfer Agreement with Rice University upon request to J.J.T. Requests will be processed within 2 weeks. Source data are provided with this paper.

## Code availability

Custom Python code for the generation of light pulsing functions with the LPA is available on GitHub (https://doi.org/10.5281/zenodo.17634803)[96].

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

## Acknowledgements

This work was supported by National Science Foundation awards CAREER 1553317 and MCB-2204402 to J.J.T. and Welch Foundation award 235019 to R.T. J.T.L. was supported by a US National Defense Science and Engineering Graduate Fellowship. D.J.H. is supported by NSF Graduate Research Fellowship 1842494. The content is solely the responsibility of the authors and does not necessarily represent the views of the funding agencies. This work was done in part using the resources of the Shared Equipment Authority (SEA) at Rice University. The authors thank Jose Avalos for suggesting the use of the Chi.Bio reactor, Chris Pennington for LC-MS guidance, and Biki Kundu for HPLC guidance.

## Author contributions

J.J.T. and J.T.L. conceived of the project. J.T.L., D.J.H., K.Y., A.R.G., and S.M.C. designed and built DNA constructs. J.T.L., D.J.H., and S.M.C. constructed the optogenetic hardware and programmed the optogenetic software. J.T.L., D.J.H., R.T., and J.J.T. designed experiments. J.T.L., D.J.H., A.G., K.Y., and J.J.K. collected data. J.T.L. and D.J.H. processed all data. J.T.L., D.J.H., and J.J.T. interpreted the results. J.T.L., D.J.H., and J.J.T. wrote the manuscript.

## Competing interests

The authors declare no competing interests.
