## [Transparent Peer Review file · Nature Communications]

A stationary phase-specific bacterial green light sensor for enhancing metabolite production

Corresponding Author: Professor Jeffrey Tabor

Version 0:

Reviewer comments:

Reviewer #1

(Remarks to the Author)

This manuscript by Lazar et al. optimized a genetically-encoded light biosensor system for practical applications. In summary, the authors addressed the malfunction issue of the previously developed CcaSR v3.0 biosensor system during the stationary phase. This malfunction was attributed to the loss of the chromophore phycocyanobilin synthesis cassette. The problem was successfully solved by integrating the cassette into the chromosome, consequently enabling the biosensor system to function effectively during the stationary phase. Expression optimization of the histidine kinase CcaS and transcriptional regulator CcaR further enhanced its performance. The biosensor system was then employed for phenylpropanoid production and showcased improved titers through dynamic light induction. Although the strategies of chromosomal integration and expression optimization are conventional, this study stands out for its systematic and logical approach, bolstered by concrete validations and well-founded reasoning. The researchers also conducted a detailed investigation into the light duty cycle and period settings. However, the following comments need to be addressed.

1. The manuscript lacks key words, which are essential for categorizing the content and aiding discoverability.
2. The manuscript introduces a novel aspect by optimizing a biosensor that functions in the stationary phase. However, the distinction between biosensors operating in the exponential growth phase and those in the stationary phase is not adequately explained. Additionally, the advantages of the biosensor system CcaSR remain unclear.
3. The engineered light-triggered biosensor system's reliance on the stationary phase, while suited for production, raises concerns about high cell density impacting light penetration and target gene regulation.
4. Control Experiments and Dynamic Regulation: Despite optimizing the CcaSR biosensor system and its application in dynamic regulation of p-coumaric acid and bidesmethoxycurcumin in *E. coli*, the manuscript lacks control experiments to substantiate the advantages of dynamic regulation. The logic requires further elucidation to underscore the strength of dynamic regulation.
5. The hypothesis suggesting that pulsatile induction leads to higher product titers is intriguing. However, the manuscript needs to delve into potential reasons for this phenomenon, particularly in the context of the observed increase in p-coumaric acid.
6. Lines 23-24. How can an optogenetic biosensor be applied in the gastrointestinal tract? How can the activation of green and red lights be achieved?
7. Lines 118-120. No such data about OD600 value was shown in Fig. S1B. The data points should be correctly labeled.
8. In Figure S1, the figure caption was needed for Fig. S1C.
9. Lines 166-168, and in Figure 2b and Figure S2C, a control group (wildtype *E. coli*) should be added to the chromosomal *ho1-psyA* cassette verification and used to prove the correct loci of the integration.
10. Line 184, it's important to explicitly state the purpose of such optimization. Clarification is needed on why there was a need to reduce the levels of CcaS and CcaR. It is recommended that Figure 3 includes a direct comparison between the system before and after optimization to illustrate the achieved results and effects.
11. Lines 184-189. How can we tell that "both proteins accumulate to higher abundance in stationary phase" from Figure S4? And why did the authors want to reduce the CcaS levels? A solid analysis should be done here.
12. Lines 228-229. The authors should prove the statement that "TAL expression is toxic in *E. coli*". Citation 41 demonstrated that the high titer of p-coumaric acid was toxic but did not contain the statement in this manuscript.
13. In both Figure 3 and Figure 4, please add a control group without the implementation of the biosensor system.

To enhance the manuscript's clarity and impact, I recommend addressing these points in detail. Providing clearer

explanations for the differentiation of biosensors in distinct growth phases, outlining the merits of the CcaSR system, discussing strategies to mitigate the high cell density impact, incorporating control experiments for dynamic regulation, and unraveling the reasons behind increased product titers due to pulsatile induction would significantly enrich the overall quality of the study.

Reviewer #2

(Remarks to the Author)

The manuscript titled "A Green Light Sensor for Enhancing Metabolite Production in Bacteria During Stationary Phase" describes the development and optimization of a CcaSRstat system that allows optogenetic control for metabolic engineering during stationary phase. The authors worked to improve the genetic context and expression determinants of the CcaSR light-switchable two-component system, resulting in a stable CcaSRstat system that operates exclusively during stationary phase, with minimal metabolic burden and low leakiness. They also integrated the *ho1-*pcyA** cassette into the chromosome to reduce metabolic load and optimized CcaS and CcaR expression levels during stationary phase. The manuscript presents the practical application of CcaSRstat in optimizing the synthesis of two phenylpropanoids in *E. coli*, achieving substantial titers of these industrially relevant compounds through CcaSRstat-mediated enzyme expression with various illumination patterns. Furthermore, these optimized conditions were successfully translated to benchtop bioreactors. While the manuscript is well-organized and informative, several aspects of the system's development and application require further discussion and revision.

Major comments:

1. Lines 72-78: It would be beneficial for the authors to provide a schematic diagram that visually explains how the two-component system CcaSR functions in regulating bacterial gene expression through the use of light.
2. Fig. 3B: The expression levels of CcaR using RBS1 and RBS2 exhibited only a slight difference. The authors should present a statistical analysis for this data and discuss the significant variation in the light response (Fig. 3C) resulting from this minor difference in CcaR expression levels. Simply stating that the "absolute CcaR level is a critical parameter for light response, regardless of the growth phase" might give the impression of a less versatile system.
3. Lines 211-217: The authors presented evidence that insufficient expression of CcaS and/or CcaR led to the characteristic of stationary phase-specificity in CcaSRstat. However, this feature significantly limits the range of achievable induction control using CcaSRstat. Induction control would then be restricted to tuning the expression levels of enzymes by adjusting light intensity and duration, without the capability to determine the initiation time of enzyme expression through a light switch. The authors should further optimize CcaSRstat by identifying the optimal expression levels of *ho1-*pcyA**, CcaS, and CcaR, enabling CcaSRstat to function effectively in both exponential and stationary growth phases.
4. Fig. 4C: In the main text, the authors stated that titers became highly variable above the intensity of $7 \mu\text{mol m}^{-2} \text{s}^{-1}$. However, these data points were not shown in the figure. Please clarify.
5. Fig. S8: CcaSRstat regulates the expression of target enzymes under the specific promoter *P_{cpcG2-172}*. This characteristic of CcaSRstat renders it more suitable for controlling the expression of individual pathway components. In the experimental design depicted in Fig. S8, the authors applied CcaSRstat to optimize a multi-enzyme pathway by expressing a single operon encoding four genes under the control of *P_{cpcG2-172}*. However, the system treated all four genes as a collective unit and was unable to modulate the expression levels of these pathway components independently. As a result, the outcomes of the optimization do not ensure optimal production and thus cannot effectively support the appropriate utilization of CcaSRstat in optimizing multi-enzyme metabolic pathways. The study's section might give readers the wrong impression that CcaSRstat can adjust the levels of several pathway components at the same time for optimization purposes. The authors should revise the main text to clarify this issue.
6. Lines 259-276: Using static light signals, $7 \mu\text{mol m}^{-2} \text{s}^{-1}$ demonstrated the highest levels of TAL-sfGFP expression (Fig. 4B-C). The authors speculated that levels of expression beyond $7 \mu\text{mol m}^{-2} \text{s}^{-1}$ might become unreliable due to potential toxicity. In the subsequent experiments using dynamic light signals (Fig. 5B), the authors hypothesized that "pulsatile induction may result in higher product titers than static induction by distributing total enzyme expression across the course of a fermentation while avoiding acute toxicity arising from high intracellular enzyme concentration at any given point in time". However, the authors directly employed the intensity of $7 \mu\text{mol m}^{-2} \text{s}^{-1}$ in the optimization using dynamic light signals, which overlooked the interplay between light intensity and pulsatile induction for reducing toxicity and increasing product titer. The authors should incorporate varying light intensities into their optimization strategy under dynamic light conditions, ensuring a more comprehensive control of target enzyme expression.
7. Lines 260-262: Dynamic light induction was shown to enhance p-coumaric acid production further. It was also indicated in the manuscript that in many cases, pulsatile induction may result in a higher product titre than static induction (line 260-262). However, the application of this dynamic light induction was not shown in the production of bidesmethoxycurcumin. Furthermore, the abstract stated that high titres of "two industrially-relevant phenylpropanoids" were achieved by combining CcaSRstat-driven enzyme expression with varied static and periodic illumination patterns. The authors should present data that illustrate the effects of dynamic illumination patterns on bidesmethoxycurcumin production.

Minor comments:

1. Lines 51-52: The authors stated “Despite their routine use in metabolic engineering, nearly all inducible promoter systems are optimized to function in exponential phase.” To support this claim, it would be helpful if the authors could provide examples along with references.
2. Lines 108-109: The phrase “the J23106 and J23108 constitutive promoters driving expression of ho1-*pcyA* and *ccaS*, respectively” seems to indicate that the J23106 promoter controlled ho1-*pcyA* expression and the J23108 promoter controlled *ccaS* expression. However, the authors mentioned in Line 171 that the J23108 promoter was responsible for driving the expression of ho1-*pcyA*. It would be helpful if the authors could clarify the precise promoters utilized for expressing ho1-*pcyA* and *ccaS* and adjust the labeling of the promoter's name in Fig. 1A.
3. Line 186: In the phrase, “we moved the corresponding expression cassette from onto a lower (~3) copy pSC101-origin plasmid.”, the authors might have missed a word (the name of the plasmid replication origin) between ‘from’ and ‘onto’.
4. Lines 186-187: To improve the logical flow, the authors should explain their decision to propose the reduction of CcaS expression as part of the optimization process.
5. Lines 186-193: Please elaborate on why two different methods were employed for optimizing the expression levels of CcaS and CcaR.
6. Line 263: Define “CcaSRstat-tal”.
7. Lines 309-321: The authors discussed several design features of the stationary phase-optimized CcaSRstat system in this paragraph. However, the key points of most stated features seem to be the same as “releasing metabolic burden or toxicity”. Please revise this part of the content to be more concise or incorporate more diverse aspects for discussion.
8. Lines 357-358: The authors stated “Future work to optimise such orthogonal light sensors for stationary phase could enable optogenetics to be used for modular optimization of multi-step metabolic pathways.” They should elaborate on the steps required to optimise orthogonal light sensors for the stationary phase.

Reviewer #3

(Remarks to the Author)

In this study, the authors have developed a stable light-regulated CcaSRstat system that imposes little metabolic burden, exhibits low leakiness and 80-fold dynamic range, and functions exclusively in the stationary phase. In addition, the stable system was further utilized for chemical production. However, before publication, several issues need to be carefully addressed.

Q1: The introduction, specifically lines 82-103, contains excessive wording and can be condensed by eliminating redundant descriptions.

Q2: In lines 113-121, the authors have observed that CcaSR v3.0 negatively affects strain growth under normal conditions (dark conditions). However, the authors must provide evidence regarding whether this negative effect also occurs under light conditions.

Q3: In lines 138-144, the authors suggest that population heterogeneity in the stationary phase may be attributable to a mutation disabling PCB biosynthesis. Nonetheless, it should be noted that using different media for culturing strains can also result in population heterogeneity. Therefore, it is recommended that the authors exclude the influence of culture medium as a contributing factor.

Q4: In lines 145-149, the authors have employed diagnostic PCR to confirm the presence of the ho1-*pcyA* cassette in both phases. It would strengthen the conclusion if the authors were to conduct protein-level experiments, such as SDS-PAGE, to validate their findings.

Q5: Although the authors have successfully optimized the CcaSRstat system for their purposes, it remains unclear whether the system is universally applicable to other types of *E. coli*. It would be beneficial if the authors could investigate and discuss the system's suitability for other strains of *E. coli*.

Q6: In the section where the authors optimize phenylpropanoid production with static light signals, they utilize the CcaSRstat system to produce p-CA and BDMC. However, no control experiments are conducted to compare the titers and OD600 levels when the CcaSRstat system is not used. Therefore, it is recommended that the authors include such control experiments to provide a baseline for comparison.

Q7: The authors focus on optimizing dynamic light induction for p-CA production in the sections related to dynamic light signals. However, they do not conduct related experiments on BDMC. This raises doubts regarding whether the CcaSRstat system is effective for multienzyme metabolic pathways under dynamic light signal conditions. It is suggested that the authors address this discrepancy and provide an explanation.

Q8: In lines 331-338, the authors should highlight specific examples detailing the advantages of the CcaSRstat system over other photogenetic systems. This would further emphasize the unique features and benefits of their proposed system.

Version 1:

Reviewer comments:

Reviewer #1

(Remarks to the Author)

The revised manuscript addressed my comments well.

Reviewer #2

(Remarks to the Author)

In the revised manuscript and rebuttal letter, the authors have carefully addressed and responded to all my comments. The revisions demonstrate substantial effort and notably improved the completeness and clarity of the work. Here, a few minor revisions are suggested for further refinement prior to publication.

Minor comments:

1. Line 206 & 209: These sentences appear to describe [Fig. 3D], not [Fig. 3B].
2. Line 213: Please also cite the figure that shows the optimal CcaR expression level in the exponential phase.
3. Figure 4D & 4G: Please add one or two sentences in the main text to describe the content of these figures.

Reviewer #1 (Remarks to the Author):

This manuscript by Lazar et al. optimized a genetically-encoded light biosensor system for practical applications. In summary, the authors addressed the malfunction issue of the previously developed CcaSR v3.0 biosensor system during the stationary phase. This malfunction was attributed to the loss of the chromophore phycocyanobilin synthesis cassette. The problem was successfully solved by integrating the cassette into the chromosome, consequently enabling the biosensor system to function effectively during the stationary phase. Expression optimization of the histidine kinase CcaS and transcriptional regulator CcaR further enhanced its performance. The biosensor system was then employed for phenylpropanoid production and showcased improved titers through dynamic light induction. Although the strategies of chromosomal integration and expression optimization are conventional, this study stands out for its systematic and logical approach, bolstered by concrete validations and well-founded reasoning. The researchers also conducted a detailed investigation into the light duty cycle and period settings. However, the following comments need to be addressed.

1. The manuscript lacks key words, which are essential for categorizing the content and aiding discoverability.

Thank you for this comment. We have added key words to the beginning of the manuscript.

2. The manuscript introduces a novel aspect by optimizing a biosensor that functions in the stationary phase. However, the distinction between biosensors operating in the exponential growth phase and those in the stationary phase is not adequately explained. Additionally, the advantages of the biosensor system CcaSR remain unclear.

We thank the reviewer for noticing these issues. We have included a description of how accumulation of transcriptional activators or repressors can alter expression of metabolic pathways during stationary phase in an unwanted fashion, and how the expression levels of such transcription factors may need to be re-optimized for stationary phase to achieve robust inducibility (lines 57-65).

We have also added descriptions of the advantages of the CcaSR system (lines 79-81) and CcaSR_{stat} specifically (lines 382-384) to the introduction and discussion, respectively.

3. The engineered light-triggered biosensor system's reliance on the stationary phase, while suited for production, raises concerns about high cell density impacting light penetration and target gene regulation.

We thank the reviewer for pointing out the challenge of light penetration in high density fermentations. Previous work has achieved light penetration for target gene regulation on the 2 L scale up to OD₆₀₀ ~ 50 (Zhao *et al.*, Nature 2018; doi: 10.1038/nature26141). Similarly, we observe no issues in our 25 mL reactor experiments (Figure 5D). That said, we agree that light penetration is an unsolved issue at larger scales and acknowledge these limitations in our discussion. In line with this point, we state that CcaSR_{stat} is most valuable as a metabolic pathway prototyping tool, allowing metabolic engineers to explore new multi-dimensional

pathway optimizations (induction timing, intensity, frequency, duty cycle) more quickly (lines 393-400). Future studies of light penetration in high density large fermentations and the engineering of new photobioreactor systems will be critical in understanding the full impact of optogenetic induction schemes for large scale fermentations.

4. Control Experiments and Dynamic Regulation: Despite optimizing the CcaSR biosensor system and its application in dynamic regulation of *p*-coumaric acid and bidesmethoxycurcumin in *E. coli*, the manuscript lacks control experiments to substantiate the advantages of dynamic regulation. The logic requires further elucidation to underscore the strength of dynamic regulation.

We thank the reviewer for this comment. To optimize *p*-coumaric acid titer, we sequentially optimize static light intensity, induction timing, and dynamic light period and duty cycle. This three-step process reveals that optimized dynamic light delivery results in higher titers than optimized static light delivery. In particular, we first find that the optimal intensity of static green light is $7 \mu\text{mol m}^{-2} \text{s}^{-1}$, resulting in a titer of $71.3 \pm 2.8 \text{ mg/L}$ (Fig 4F, reproduced below).

Next, we grow bacteria in deactivating red light for 0-12 h and switch to the $7 \mu\text{mol m}^{-2} \text{s}^{-1}$ static green light at 0, 2, 4, 6, 8, or 12 h into the culture growth. The results demonstrate that a 6 h induction time ($\text{OD} \sim 0.56 \pm 0.02$), increases *p*-CA titer to a new optimum of $81.7 \pm 5.5 \text{ mg/L}$ (Fig. S18B-D, reproduced below)

Next, we optimize the period and duty cycle of a periodic pulse of $7 \mu\text{mol m}^{-2} \text{s}^{-1}$ green light applied at a 6 h induction time. This third level optimization further increases the titer to $113.1 \pm 39.7 \text{ mg/L}$ at a period of 10 minutes and a duty cycle of 85% (Fig. 5B, reproduced below).

Finally, we compare our optimized static and dynamic light treatments in scaled-up 25 mL Chi.Bio reactors (compared to the original 0.5 mL Light Plate Apparatuses). We find that the dynamic light program still outperforms the static program at this larger reactor scale (final titers for the pulse and static programs are $87.8 \pm 12.4 \text{ mg/L}$ and $34.8 \pm 5.2 \text{ mg/L}$, respectively), producing larger amounts of *p*-CA further into the fermentation (Fig. 5D, reproduced below).

Overall, we use *p*-CA titer to directly compare optimized dynamic and optimized static light treatment at two different fermentation scales. We believe these experiments address the reviewer's concern.

5. The hypothesis suggesting that pulsatile induction leads to higher product titers is intriguing. However, the manuscript needs to delve into potential reasons for this phenomenon, particularly in the context of the observed increase in *p*-coumaric acid.

We thank the reviewer for the comment. We have revised the main text (lines 308-318) and added Fig. S20 (reproduced below) to discuss the reasons for the increase in *p*-CA production with pulsatile light.

We find that the optimized static light signal results in lower densities of CcaSR_{stat}-*tal* cultures than the optimized dynamic light signal (Fig. S20A). Notably, this effect is absent in non-engineered (WT) *E. coli*. Together, these results indicate that static light treatment is more toxic to *E. coli* than pulsatile light treatment due to TAL expression patterns. Strikingly, using the strain CcaSR_{stat}-*tal*-*sfGFP*, we find that TAL-sfGFP expression levels are identical in static and pulsatile light conditions over the large majority of a three-day LPA fermentation (Fig. S20B).

This result rules out TAL expression level as the culprit. Thus, we believe that pulsatile light reduces the toxicity of TAL expression by periodically allowing cells time to regenerate L-tyrosine and liberating gene expression machinery (RNAPs, ribosomes) to make other housekeeping gene products. Ultimately, this lowered toxicity results in a higher saturating OD₆₀₀ and thus more biomass to convert into the desired *p*-coumaric acid product.

6. Lines 23-24. How can an optogenetic biosensor be applied in the gastrointestinal tract? How can the activation of green and red lights be achieved?

Previously, our group utilized CcaSR to induce production of a longevity-enhancing exopolysaccharide from engineered *E. coli* in the gastrointestinal tract of live *C. elegans* nematodes (Hartsough *et al.* *eLife* 2020, DOI: 10.7554/eLife.56849). These roundworms are optically transparent, enabling transmission of green and red light into the gut environment. More recently, another group used CcaSR to induce expression of a Pb-detoxifying enzyme in *E. coli* residing in the gastrointestinal tract of *D. melanogaster* fruit flies (Wang *et al.* *ACS Synthetic Biology* 2024, DOI: 10.1021/acssynbio.4c00409). However, these technologies do not currently translate into larger animals. The reviewer's comment has made us realize that this statement was thus a distraction in the abstract, and we have removed it. We have included a discussion of this possible work in transparent hosts in the discussion.

7. Lines 118-120. No such data about OD600 value was shown in Fig. S1B. The data points should be correctly labeled.

We apologize for the mistake in calling the incorrect figure and have adjusted the text to call the proper growth curve.

8. In Figure S1, the figure caption was needed for Fig. S1C.

We have added a figure description for Fig. S1D-E (an expanded version of the original Fig. S1C).

9. Lines 166-168, and in Figure 2b and Figure S2C, a control group (wildtype *E. coli*) should be added to the chromosomal *ho1-psyA* cassette verification and used to prove the correct loci of the integration.

We have added a wild-type *E. coli* control (BW29655), which we now show in a new Fig. S4 (reproduced below).

10. Line 184, it's important to explicitly state the purpose of such optimization. Clarification is needed on why there was a need to reduce the levels of *CcaS* and *CcaR*. It is recommended that Figure 3 includes a direct comparison between the system before and after optimization to illustrate the achieved results and effects.

Thank you for this helpful comment. We have created a *CcaSR* v3.0 strain with optimized chromosomal *ho1-pcyA* and characterized its light response in exponential and stationary phases (Fig. 3B, reproduced below). Because this system still has optimized expression of *ccaS* and *ccaR* for exponential phase, we find that it performs optimally in exponential phase. Due to accumulation of the components, the dynamic range drops significantly in stationary phase, as the red light response becomes exceedingly leaky. This result motivates the need to re-optimize expression of *ccaS* and *ccaR* or stationary phase. We believe this new experiment improves the manuscript as the reviewer indicates.

11. Lines 184-189. How can we tell that “both proteins accumulate to higher abundance in stationary phase” from Figure S4? And why did the authors want to reduce the CcaS levels? A solid analysis should be done here.

We thank the reviewer for the suggestion.

Our past work has demonstrated that CcaS expression must reside in a specific range for proper CcaSR signal transduction (Schmidl *et al. ACS Synthetic Biology* 2014, DOI: 10.1021/sb500273n; Castillo-Hair *et al. Nature Communications* 2019, DOI: 10.1038/s41467-019-10906-6). Considering we find that stationary phase CcaS levels increase compared to their previously-optimized exponential phase levels, we aimed to reduce stationary phase CcaS levels to be closer to that of their optimum exponential phase levels.

We have reformatted Fig. S5 (originally Fig. S4) and included additional analyses to the caption to motivate our characterization and engineering decisions. In the caption, we now provide the specific mean MEFL values to provide insight into how CcaS-sfGFP/CcaR-sfGFP increase in fluorescence from exponential to stationary phase and now describe the need for optimizing CcaS levels.

12. Lines 228-229. The authors should prove the statement that “TAL expression is toxic in *E. coli*”. Citation 41 demonstrated that the high titer of *p*-coumaric acid was toxic but did not contain the statement in this manuscript.

We apologize for this error. We have updated the text to read “However, TAL-mediated *p*-CA production is toxic in *E. coli* and results in growth defects and lower total fermentation biomass” (lines 251-252).

13. In both Figure 3 and Figure 4, please add a control group without the implementation of the biosensor system.

We thank the reviewer for this comment. We have included a Δ CcaSR control in Fig. 3D (reproduced below)

In addition, we have included a control for a strain with only the output TAL-sfGFP plasmid in Fig. 4D (reproduced below):

Finally, we have also included controls for *p*-coumaric acid production with both the wild-type *E. coli* strain and a control of a strain with only the output TAL plasmid in Fig. 4G.

To enhance the manuscript's clarity and impact, I recommend addressing these points in detail. Providing clearer explanations for the differentiation of biosensors in distinct growth phases, outlining the merits of the CcaSR system, discussing strategies to mitigate the high cell density impact, incorporating control experiments for dynamic regulation, and unraveling the reasons behind increased product titers due to pulsatile induction would significantly enrich the overall quality of the study.

Thank you for this helpful guidance. We believe we have addressed this comment in the edits above.

Reviewer #2 (Remarks to the Author):

The manuscript titled "A Green Light Sensor for Enhancing Metabolite Production in Bacteria During Stationary Phase" describes the development and optimization of a CcaSRstat system that allows optogenetic control for metabolic engineering during stationary phase. The authors worked to improve the genetic context and expression determinants of the CcaSR light-switchable two-component system, resulting in a stable CcaSRstat system that operates exclusively during stationary phase, with minimal metabolic burden and low leakiness. They also integrated the *ho1-pcyA* cassette into the chromosome to reduce metabolic load and optimized CcaS and CcaR expression levels during stationary phase. The manuscript presents the practical application of CcaSRstat in optimizing the synthesis of two phenylpropanoids in *E. coli*, achieving substantial titers of these industrially relevant compounds through CcaSRstat-mediated enzyme expression with various illumination patterns. Furthermore, these optimized conditions were successfully translated to benchtop bioreactors. While the manuscript is well-organized and informative, several aspects of the system's development and application require further discussion and revision.

Major comments:

1. Lines 72-78: It would be beneficial for the authors to provide a schematic diagram that visually explains how the two-component system CcaSR functions in regulating bacterial gene expression through the use of light.

We thank the reviewer for the suggestion. In response, we have modified Fig. 1A (reproduced below) to show how this two-component system regulates gene expression with light. We have moved the simplified representation of CcaSR, focusing solely on the role of the different genetic components, to Fig. S1A.

2. Fig. 3B: The expression levels of CcaR using RBS1 and RBS2 exhibited only a slight difference. The authors should present a statistical analysis for this data and discuss the significant variation in the light response (Fig. 3C) resulting from this minor difference in CcaR expression levels. Simply stating that the "absolute CcaR level is a critical parameter for light response, regardless of the growth phase" might give the impression of a less versatile system.

Thank you for this comment. In investigating this question, we found that the *ccaR-sfgfp* fusion that we had used to characterize *ccaR* expression level encoded an unintended second start codon at the beginning of the *sfgfp* gene, allowing for expression of sfGFP proteins that may have contaminated our signal. To address this issue, we removed the unwanted start codon from all of our *ccaR-sfgfp* fusions and re-measured the expression of each in stationary (Fig. 3C, reproduced below) and exponential (Fig. S6B, below).

Fig. 3C

Fig. S6B

Additionally, we have added the results of a statistical analysis to both figures and the text demonstrating a significant increase in the expression of CcaR-sfGFP with sRBS2 compared sRBS1 in exponential ($p = 3.4 \times 10^{-4}$, student's two-tailed T-test) and stationary phase ($p = 3.3 \times 10^{-5}$, student's two-tailed T-test).

Finally, we also fixed the *sfgfp* RBS problem in our measurement of *ccaR* expression in the original CcaSR v3.0 system (Fig. S5E).

3. Lines 211-217: The authors presented evidence that insufficient expression of CcaS and/or CcaR led to the characteristic of stationary phase-specificity in CcaSR_{stat}. However, this feature significantly limits the range of achievable induction control using CcaSR_{stat}. Induction control would then be restricted to tuning the expression levels of enzymes by adjusting light intensity and duration, without the capability to determine the initiation time of enzyme expression through a light switch. The authors should further optimize CcaSR_{stat} by identifying the optimal expression levels of *ho1-*pcyA**, CcaS, and CcaR, enabling CcaSR_{stat} to function effectively in both exponential and stationary growth phases.

Thank you for this helpful comment. We now provide an alternative strain (Fig. S8, reproduced below) where we increase *ccaS* gene expression relative to CcaSR_{stat} by placing the expression cassette on a higher copy plasmid (p15a). This strain enables optical control of gene expression in both exponential and stationary phases (Fig. S8B, C).

4. Fig. 4C: In the main text, the authors stated that titers became highly variable above the intensity of $7 \mu\text{mol m}^{-2} \text{s}^{-1}$. However, these data points were not shown in the figure. Please clarify.

We apologize for not including these data points in the figure. We now include the data points tested for the 4 light intensities tested in the toxic regime in (Fig. 4F, below).

5. Fig. S8: CcaSRstat regulates the expression of target enzymes under the specific promoter PcpG2-172. This characteristic of CcaSRstat renders it more suitable for controlling the expression of individual pathway components. In the experimental design depicted in Fig. S8, the authors applied CcaSRstat to optimize a multi-enzyme pathway by expressing a single operon encoding four genes under the control of PcpG2-172. However, the system treated all four genes as a collective unit and was unable to modulate the expression levels of these pathway components independently. As a result, the outcomes of the optimization do not ensure optimal

production and thus cannot effectively support the appropriate utilization of CcaSR_{stat} in optimizing multi-enzyme metabolic pathways. The study's section might give readers the wrong impression that CcaSR_{stat} can adjust the levels of several pathway components at the same time for optimization purposes. The authors should revise the main text to clarify this issue.

We apologize for this confusing wording. As the reviewer notes, in our original manuscript submission we reported on the use of CcaSR_{stat} to optimize the expression level of a four enzyme pathway leading to the production of bidesmethoxycurcumin (BDMC) (formerly Fig. S8). During our revision process, we encountered difficulties in reproducibly quantifying BDMC levels. As a result, we removed the BDMC pathway experiments from the manuscript and replaced them with new experiments wherein we use CcaSR_{stat} to optimize the production of plant-derived betaxanthin compounds via a three-enzyme pathway in *E. coli* (Figs. S10-S16). In these new experiments, we are able to robustly detect the pathway products betalamic acid (Fig. S11), dopaxanthin (Fig. S12), and portulacaxanthin II (Fig. S13) via LC Q-TOF MS (Fig. S11-S13) as well as culture fluorescence (Fig. S14) enabling higher-throughput optogenetic experiments. We believe that these new betaxanthin results are stronger and better demonstrate the utility of our method for optimizing multi-enzyme pathways.

That said, the reviewer's original comment applies to these new experiments. In particular, we used CcaSR_{stat} to optimize the expression of three enzymes encoded in a single operon (i.e. co-transcribed from the same promoter). To address the reviewer's comment, we have replaced the original text with a new section describing betaxanthin optimization, being careful to avoid wording that may confuse readers into thinking that CcaSR_{stat} can be independently used to optimize different steps of a multi-enzyme pathway (lines 269-286). In addition, we include a section in the discussion about the limitations of CcaSR_{stat} for multi-enzyme pathways and how multiplexing light sensors with different wavelengths will be necessary to optimize each module in multi-enzyme pathways (lines 401-422).

6. Lines 259-276: Using static light signals, 7 $\mu\text{mol m}^{-2} \text{s}^{-1}$ demonstrated the highest levels of TAL-sfGFP expression (Fig. 4B-C). The authors speculated that levels of expression beyond 7 $\mu\text{mol m}^{-2} \text{s}^{-1}$ might become unreliable due to potential toxicity. In the subsequent experiments using dynamic light signals (Fig. 5B), the authors hypothesized that "pulsatile induction may result in higher product titers than static induction by distributing total enzyme expression across the course of a fermentation while avoiding acute toxicity arising from high intracellular enzyme concentration at any given point in time". However, the authors directly employed the intensity of 7 $\mu\text{mol m}^{-2} \text{s}^{-1}$ in the optimization using dynamic light signals, which overlooked the interplay between light intensity and pulsatile induction for reducing toxicity and increasing product titer. The authors should incorporate varying light intensities into their optimization strategy under dynamic light conditions, ensuring a more comprehensive control of target enzyme expression.

We thank the reviewer for this suggestion. To address this comment, we tested multiple green light intensities both above and below the optimal intensity of 7 $\mu\text{mol m}^{-2} \text{s}^{-1}$. We included the results of this experiment in Fig. S21 (reproduced below).

We found that below the light intensity threshold resulting in toxicity, the optimized pulsatile light program does not increase *p*-CA production relative to the static program, possibly due to insufficient photon flux. However, for intensities in the toxic regime of TAL induction, the pulse program results in an increase in *p*-CA titer compared to the static program. These results suggest that the pulsatile light program generally increases *p*-CA production in toxic regimes. We describe this experiment on lines 303-304 of the main text. This result could apply to other toxic metabolic enzymes as well.

7. Lines 260-262: Dynamic light induction was shown to enhance *p*-coumaric acid production further. It was also indicated in the manuscript that in many cases, pulsatile induction may result in a higher product titre than static induction (line 260-262). However, the application of this dynamic light induction was not shown in the production of bisdemethoxycurcumin. Furthermore, the abstract stated that high titres of “two industrially-relevant phenylpropanoids” were achieved by combining *CcaSR*_{stat}-driven enzyme expression with varied static and periodic illumination patterns. The authors should present data that illustrate the effects of dynamic illumination patterns on bisdemethoxycurcumin production.

We thank the reviewer for this comment. As mentioned above, we have replaced the BDMC pathway optimization with betaxanthin pathway optimization experiments. In particular, we use static light treatment to optimize betaxanthin pathway productivity (Fig. S15), and show that betaxanthins are primarily produced in stationary phase, consistent with *CcaSR*_{stat} function (Fig. S16). We also demonstrate that dynamic optimization of this pathway fails to improve betaxanthin titer over static illumination, as measured through a fluorescence assay (Fig. S22), likely due to the aforementioned limitations of *CcaSR*_{stat} in controlling only one promoter, which are described in the main text.

Minor comments:

1. Lines 51-52: The authors stated “Despite their routine use in metabolic engineering, nearly all inducible promoter systems are optimized to function in exponential phase.” To support this claim, it would be helpful if the authors could provide examples along with references.

We have added references for this claim.

2. Lines 108-109: The phrase “the J23106 and J23108 constitutive promoters driving expression

of *ho1-pcyA* and *ccaS*, respectively” seems to indicate that the J23106 promoter controlled *ho1-pcyA* expression and the J23108 promoter controlled *ccaS* expression. However, the authors mentioned in Line 171 that the J23108 promoter was responsible for driving the expression of *ho1-pcyA*. It would be helpful if the authors could clarify the precise promoters utilized for expressing *ho1-pcyA* and *ccaS* and adjust the labeling of the promoter's name in Fig. 1A.

We apologize for the error. The labels in Fig. 1A are correct, and we have updated the main text accordingly (lines 108-109).

3. Line 186: In the phrase, “we moved the corresponding expression cassette from onto a lower (~3) copy pSC101-origin plasmid.”, the authors might have missed a word (the name of the plasmid replication origin) between ‘from’ and ‘onto’.

Thank you. We have fixed the grammatical error.

4. Lines 186-187: To improve the logical flow, the authors should explain their decision to propose the reduction of *CcaS* expression as part of the optimization process.

We have included text explaining the need to reduce stationary phase *CcaS* levels to be closer to optimal exponential phase (lines 195-196).

5. Lines 186-193: Please elaborate on why two different methods were employed for optimizing the expression levels of *CcaS* and *CcaR*.

Our previous work has shown that sensor histidine kinases can be expressed over a wide range while response regulators must be expressed in a relatively narrow range for optimal two-component system function. Thus, we performed a relatively coarse optimization of *ccaS* expression levels via plasmid copy number, and used a finer RBS library-mediated optimization to tune *ccaR*. We have updated the text to reflect this point (lines 198-200).

6. Line 263: Define “*CcaSR_{stat}-tal*”.

We have now defined *CcaSR_{stat}-tal* in the main text as “the strain where *CcaSR_{stat}* controls expression of *tal*”.

7. Lines 309-321: The authors discussed several design features of the stationary phase-optimized *CcaSR_{stat}* system in this paragraph. However, the key points of most stated features seem to be the same as “releasing metabolic burden or toxicity”. Please revise this part of the content to be more concise or incorporate more diverse aspects for discussion.

Thank you for this comment. We have edited this paragraph according to the reviewer’s suggestion (lines 358-366).

8. Lines 357-358: The authors stated “Future work to optimise such orthogonal light sensors for stationary phase could enable optogenetics to be used for modular optimization of multi-step metabolic pathways.” They should elaborate on the steps required to optimise orthogonal light

sensors for the stationary phase.

We have included detailed information on the steps required to optimize these orthogonal light sensors for stationary phase and co-expressing them in a single cell for multiplexed optogenetic control of multi-step enzymatic pathways (lines 401-422).

Reviewer #3 (Remarks to the Author):

In this study, the authors have developed a stable light-regulated CcaSRstat system that imposes little metabolic burden, exhibits low leakiness and 80-fold dynamic range, and functions exclusively in the stationary phase. In addition, the stable system was further utilized for chemical production. However, before publication, several issues need to be carefully addressed.

Q1: The introduction, specifically lines 82-103, contains excessive wording and can be condensed by eliminating redundant descriptions.

Thank you for this comment. We have condensed this section in the revised manuscript (lines 91-103)

Q2: In lines 113-121, the authors have observed that CcaSR v3.0 negatively affects strain growth under normal conditions (dark conditions). However, the authors must provide evidence regarding whether this negative effect also occurs under light conditions.

We thank the reviewer for the suggested experiment. We have measured how red/green light affects growth of CcaSR v3.0 and wildtype (WT) *E. coli*, and have included the plot in Fig. S1C (reproduced below). These data show that red and green light have no additional effect beyond that of CcaSR v3.0 on growth.

Q3: In lines 138-144, the authors suggest that population heterogeneity in the stationary phase may be attributable to a mutation disabling PCB biosynthesis. Nonetheless, it should be noted that using different media for culturing strains can also result in population heterogeneity. Therefore, it is recommended that the authors exclude the influence of culture medium as a contributing factor.

We thank the reviewer for the suggestion. Our previously reported experiments were performed in defined M9 media. In response to this comment, we additionally tested two rich media formulations (LB and 2xYT) with CcaSR v3.0 in exponential and stationary phase (Fig. S1G-J, and below).

As can be seen in the figures, we still observe stationary-phase dependent heterogeneity in red and green light. We believe these results address the reviewer's comment.

We also note that with both LB and 2xYT media formulations, the green light response of CcaSR v3.0 additionally becomes more heterogenous than in M9 (Fig. S1D-E, shown below for comparison). This data agrees with the reviewer's comment that media formulation can affect heterogeneity.

Q4: In lines 145-149, the authors have employed diagnostic PCR to confirm the presence of the ho1-*pcyA* cassette in both phases. It would strengthen the conclusion if the authors were to conduct protein-level experiments, such as SDS-PAGE, to validate their findings.

We thank the reviewer for the suggestion. We performed SDS-PAGE analysis on the whole cell lysate of wild-type cells, cells with plasmid-based *hoI-pcyA* expression, and cells with chromosomally-optimized *hoI-pcyA* expression in exponential (E) and stationary (S) phase (below)

The expected size of both the HoI and PcyA proteins is ~27kDa. Unfortunately, we could not detect expression of these enzymes in this assay due to the fact that they are not strongly overexpressed, and the fact that they overlap with several cellular proteins of similar size. Despite this result, we believe our diagnostic PCR analysis, sequencing of the plasmid-borne mutations, and Cph1(Y176H) PCB biosensor experiments provide strong evidence of the loss and engineering of stationary phase PCB biosynthesis.

Q5: Although the authors have successfully optimized the CcaSR_{stat} system for their purposes, it remains unclear whether the system is universally applicable to other types of *E. coli*. It would be beneficial if the authors could investigate and discuss the system's suitability for other strains of *E. coli*.

We thank the reviewer for this thoughtful comment. To address it, we separately integrated our optimized chromosomal *hoI-pcyA* cassette into the *torT* and *yhbC* loci of the well-studied probiotic strain *E. coli* Nissle 1917 (EcN) (Fig. S9, reproduced below). We then transformed the three CcaSR_{stat} plasmids into each of these two modified EcN strains (Fig. S9A). This EcN CcaSR_{stat} exhibits low leakiness in red light, strong gene expression activation in green (Fig. S9B), and specificity for stationary phase (Fig. S9C-F), recapitulating its performance in *E. coli* BW29655. As BW29655 is a laboratory-adapted strain and EcN is a non-domesticated gut-adapted strain, we believe CcaSR_{stat} is likely to work in a wide variety of *E. coli* strains.

Q6: In the section where the authors optimize phenylpropanoid production with static light signals, they utilize the CcaSR_{stat} system to produce p-CA and BDMC. However, no control experiments are conducted to compare the titers and OD600 levels when the CcaSR_{stat} system is not used. Therefore, it is recommended that the authors include such control experiments to provide a baseline for comparison.

We thank the author for suggesting the comparison between strains with and without the CcaSR_{stat} system. For p-CA, we have added **Fig. 4G** to demonstrate that in a strain with the output *tal* plasmid but lacking CcaSR_{stat}, no p-CA is detectable.

Due to concerns about the reproducibility of our BDMC data described above, we have replaced our BDMC pathway experiments with experiments optimizing the multi-enzyme betaxanthin pathway (**Fig. S10**). We demonstrate optimization of production of betaxanthins using static light (**Fig. S15**), as measured through a culture fluorescence assay and LC Q-TOF MS (**Fig. S11-S14**). The LC Q-TOF MS data (**Fig. S11-S13**) show that no betaxanthins are detectable when the CcaSR_{stat} system is absent from the strain, which we believe addresses the reviewer's concern.

Q7: The authors focus on optimizing dynamic light induction for p-CA production in the sections related to dynamic light signals. However, they do not conduct related experiments on BDMC. This raises doubts regarding whether the CcaSR_{stat} system is effective for multienzyme

metabolic pathways under dynamic light signal conditions. It is suggested that the authors address this discrepancy and provide an explanation.

This is an important comment. As discussed above, we switched from optimizing BDMC production to optimizing betaxanthin production due to concerns about the reproducibility of the BDMC data. We show that optimization of the betaxanthin pathway with static light signals can achieve increased titer, as measured by betaxanthin fluorescence (**Fig. S15**). However, optimization of the pathway using dynamic light signals failed to identify conditions in which fluorescence was increased with statistical significance relative to the best static light condition (**Fig. S22**). This result highlights limitations of CcaSR_{stat} in optimizing multi-enzyme pathways, a result which we believe is shared by any single inducible promoter system. We highlight this limitation in the discussion section and discuss future work that could address it (lines 407-427).

Q8: In lines 331-338, the authors should highlight specific examples detailing the advantages of the CcaSR_{stat} system over other photogenetic systems. This would further emphasize the unique features and benefits of their proposed system.

We thank the author for the suggestion. We have added descriptions of the advantages of CcaSR in general (lines 79-81) and CcaSR_{stat} in particular (lines 382-384) to the introduction and discussion, respectively.

Point by point reviewer response

Reviewer #1 (Remarks to the Author):

The revised manuscript addressed my comments well.

Thank you for your careful evaluation of our paper.

Reviewer #2 (Remarks to the Author):

In the revised manuscript and rebuttal letter, the authors have carefully addressed and responded to all my comments. The revisions demonstrate substantial effort and notably improved the completeness and clarity of the work. Here, a few minor revisions are suggested for further refinement prior to publication.

Minor comments:

1. Line 206 & 209: These sentences appear to describe [Fig. 3D], not [Fig. 3B].

Thank you. We have corrected this mistake.

2. Line 213: Please also cite the figure that shows the optimal CcaR expression level in the exponential phase.

Done.

3. Figure 4D & 4G: Please add one or two sentences in the main text to describe the content of these figures.

Thank you. We have added one sentence for each of these figure panels.